# INTEPRETING & IMPROVING PRETRAINED LANGUAGE MODELS: A PROBABILISTIC CONCEPTUAL APPROACH

## ABSTRACT

Pretrained Language Models (PLMs) such as BERT and its variants have achieved remarkable success in natural language processing. To date, the interpretability of PLMs has primarily relied on the attention weights in their self-attention layers. However, these attention weights only provide word-level interpretations, failing to capture higher-level structures, and are therefore lacking in readability and intuitiveness. In this paper, we propose a hierarchical Bayesian deep learning model, dubbed continuous latent Dirichlet allocation (CLDA), to go beyond word-level interpretations and provide concept-level interpretations. Our CLDA is compatible with any attention-based PLMs and can work as either (1) an interpreter which interprets model predictions at the concept level without any performance sacrifice or (2) a regulator which is jointly trained with PLMs during finetuning to further improve performance. Experimental results on various benchmark datasets show that our approach can successfully provide conceptual interpretation and performance improvement for state-of-the-art PLMs.

## 1 INTRODUCTION

Pretrained language models (PLMs) such as BERT Devlin et al. (2018) and its variants Lan et al. (2019); Liu et al. (2019); He et al. (2021) have achieved remarkable success in natural language processing. These PLMs are usually large attention-based neural networks that follow a pretrain-finetune paradigm, where models are first pretrained on large datasets and then finetuned for a specific task. As with any machine learning models, interpretability in PLMs has always been a desideratum, especially in decision-critical applications (e.g., healthcare).

To date, the interpretability of PLMs has primarily relied on the attention weights in their self-attention layers. However, these attention weights only provide raw word-level importance scores as interpretations. Such low-level interpretations fail to capture higher-level semantic structures, and are therefore lacking in readability, intuitiveness and stability. For example, low-level interpretations often fail to capture influence of similar words to predictions, leading to unstable or even unreasonable explanations (see Sec. 4.2 for details).

In this paper, we make an attempt to go beyond word-level attention and interpret PLM predictions at the concept (topic) level. Such higher-level semantic interpretations are complementary to word-level importance scores and tend to more readable and intuitive. The core of our idea is to treat a PLM's contextual word embeddings (and their corresponding attention weights) as observed variables and build a probabilistic generative model to automatically infer the higher-level semantic structures (e.g., concepts or topics) from these embeddings and attention weights, thereby interpreting the PLM's predictions at the concept level.

Specifically, we propose a class of hierarchical Bayesian deep learning models, dubbed continuous latent Dirichlet allocation (CLDA), to (1) discover concepts (topics) from contextual word embeddings and attention weights in PLMs and (2) interpret individual model predictions using these concepts. It is worth noting that CLDA is 'continuous' because it treats attention weights as *continuous-value* word counts and models contextual word embeddings with *continuous-value* entries; this is in stark contrast to typical latent Dirichlet allocation Blei et al. (2003) that can only handle bag-of-words (both words and word counts are discrete values). Our CLDA is compatible with any attention-based PLMs and can work as either an interpreter, which interprets model predictions at the concept level without

any performance sacrifice, or a regulator, which is jointly trained with PLMs during finetuning to further improve performance. Our contributions are as follows:

- We propose a novel class of models, CLDA, to go beyond word-level interpretations and interpret PLM predictions at the concept level, thereby improving readability and intuitiveness.
- Our CLDA is compatible with any attention-based PLMs and can work as either an interpreter, which interprets model predictions without performance sacrifice, or a regulator, which is jointly trained with PLMs during finetuning to further improve performance.
- We provide empirical results across various benchmark datasets which show that CLDA can successfully interpret predictions from various PLM variants at the concept level and improve PLMs' performance when working as a regulator.

## 2 RELATED WORK

**Pretrained Language Models.** Pretrained language models are large attention-based neural networks that follow a pretrain-finetune paradigm. Usually they are first pretrained on large datasets in a self-supervised manner and then finetuned for a specific downstream task. BERT Devlin et al. (2018) is a pioneering PLM that has shown impressive performance across multifple downstream tasks. Following BERT, there have been variants, such as Albert Lan et al. (2019), DistilBERT Sanh et al. (2019), and Tinybert Jiao et al. (2019), that achieve performance comparable to BERT with fewer parameters. Other variants such as RoBERTa Liu et al. (2019) and BART Lewis et al. (2019) improve the performance using more sophisticated training schemes for the masked language modeling learning objective. More recently, there have also been BERT variants that design different self-supervised learning objectives to achieve better performance; examples include DeBERTa He et al. (2021), Electra Clark et al. (2020), and XLNet Yang et al. (2019). While these PLMs naturally provide attention weights for each word to intepret model predictions, such low-level interpretations fail to capture higher-level semantic structures, and are therefore lacking in readability and intuitiveness. In contrast, our CLDA goes beyond word-level attention and interpret PLM predictions at the concept (topic) level. These higher-level semantic interpretations are complementary to word-level importance scores and tend to more readable and intuitive.

**Topic Models.** Our work is also related to topic models Blei (2012); Blei et al. (2003), which typically build upon latent Dirichlet allocation (LDA) Blei et al. (2003). Topic models takes the (discrete) bag-of-words representations of the documents (i.e., vocabulary-length vectors that count word occurrences) as input, discover hidden topics from them during training, and infer the topic proportion vector for each document during inference Blei et al. (2003); Blei & Lafferty (2006); Wang et al. (2012); Chang & Blei (2009). Besides these 'shallow' topic models, there has been recent work that employs 'deep' neural networks to learn topic models more efficiently Card et al. (2017); Xing et al. (2017); Peinelt et al. (2020), using techniques such as amortized variational inference. There is also work that improves upon traditional topic models by either leveraging word similarity as a regularizer for topic-word distributions Das et al. (2015); Batmanghelich et al. (2016) or including word embeddings into the generative process Hu et al. (2012); Dieng et al. (2020); Bunk & Krestel (2018). Here we note several key differences between our CLDA and the methods above. (1) These methods focus on learning topic models *from scratch* given a collection of raw documents, while our CLDA learns topic models directly *from the latent representations inside PLMs*. (2) They assume word representations are *static* (i.e., the representation of a word remains the same across different documents), while PLMs' word representations are *contextual* (i.e., the representation of a word varies across different documents according to context). In contrast, our CLDA does not have such an assumption. (3) They assume word counts are discrete numbers, which is not applicable to PLMs where each word has a continuous-valued (or real-valued) word count (i.e., its attention weight). In contrast, our CLDA naturally handles continuous-valued word counts from PLMs in a differentiable manner to enable end-to-end training. Therefore **these prior methods are not applicable to PLMs**.

## 3 METHODS

In this section, we formalize the problem of conceptual interpretation of PLMs, and describe our methods for addressing this problem.

**Problem Setting and Notation.** We consider a corpus of $M$ documents, where the $m$'th document contains $L_m$ words, and a PLM $f(\mathcal{D}_m)$, which takes as input the document $m$ (denoted as $\mathcal{D}_m$) with $L_m$ words and outputs (1) a CLS embedding $\mathbf{c}_m \in \mathbb{R}^d$, (2) $L_m$ contextual word embeddings $\mathbf{e}_m \triangleq [\mathbf{e}_{mj}]_{j=1}^{L_m}$, and (3) the attention weights $\mathbf{a}_m^{(h)} \triangleq [a_{mj}^{(h)}]_{j=1}^{L_m}$ between each word and the last-layer CLS token, where $h$ denotes the $h$'th attention head. We denote the average attention weight over H heads as $a_{mj} = \frac{1}{H}\sum_{h=1}^{H} a_{mj}^{(h)}$ and correspondingly $\mathbf{a}_m \triangleq [a_{mj}]_{j=1}^{L_m}$ (see the PLM at the bottom of Fig. 1). In PLMs, these last-layer CLS embeddings are used as document-level representations for downstream tasks (e.g., document classification). Furthermore, our CLDA assumes $K$ concepts (topics) for the corpus. For document $m$, our CLDA interpreter tries to infer a concept distribution vector $\boldsymbol{\theta}_m \in \mathbb{R}^K$ (also known as the topic proportion in topic models) for the whole document and a concept distribution vector $\phi_{mj} = [\phi_{mji}]_{i=1}^K \in \mathbb{R}^K$ for word $j$ in document $m$. In our continuous embedding space, the $i$'th concept is represented by a Gaussian distribution, $\mathcal{N}(\boldsymbol{\mu}_i, \boldsymbol{\Sigma}_i)$, of contextual word embeddings. The goal is to interpret PLMs' predictions *at the concept level* using the inferred document-level concept vector $\boldsymbol{\theta}_m$, word-level concept vector $\phi_{mj}$, and the learned embedding distributions $\{\mathcal{N}(\boldsymbol{\mu}_i, \boldsymbol{\Sigma}_i)\}_{i=1}^K$ for each concept (see Sec. 4.2 for more detailed descriptions and visualizations).

## 3.1 Continuous Latent Dirichlet Allocation

**Method Overview.** Different from *static* word embeddings Mikolov et al. (2013) and topic models, PLMs produce *contextual* word embeddings with continuous-value entries $[\mathbf{e}_{mj}]_{j=1}^{L_m}$ and more importantly, associate each word embedding with a continuous-value attention weight $[a_{mj}]_{j=1}^{L_m}$; therefore this brings unique challenges.

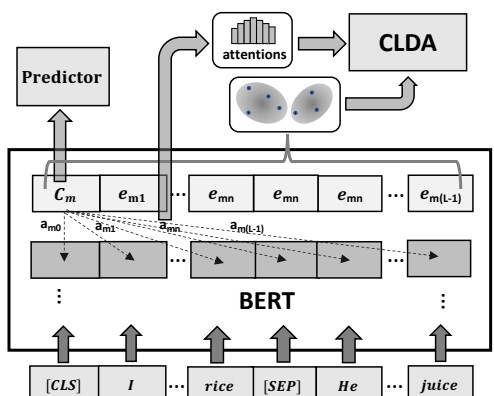

Figure 1: Overview of our CLDA framework.

To effectively discover latent concept structures learned by PLMs at the dataset level and interpret PLM predictions at the data-instance level, our CLDA treats both the contextual word embeddings and their associated attention weights as observations to learn a probabilistic generative model of these observations, as shown in Fig. 1. The key idea is to use the attention weights from PLMs to compute a virtual continuous count for each word, and model the contextual word embedding distributions with Gaussian mixtures. The generative process of CLDA is as follows (we mark key differences from LDA in blue and show the corresponding graphical model in Fig. 2):

1. For each document $m, 1 \le m \le M$,
    (a) Draw the document-level concept distribution vector $\boldsymbol{\theta}_m \sim \text{Dirichlet}(\boldsymbol{\alpha})$
    (b) For each word $j, 1 \le j \le L_m$,
        i. Draw the word-level concept index $z_{mj} \sim \text{Categorical}(\boldsymbol{\theta}_m)$
        ii. With a continuous word count $w_{mj} \in \mathbb{R}$ from the PLM's attention weights,
            A. Draw the contextual word embedding of the PLM from the corresponding Gaussian component $\mathbf{e}_{mj} \sim \mathcal{N}(\boldsymbol{\mu}_{z_{mj}}, \boldsymbol{\Sigma}_{z_{mj}})$

Given the generative process above, discovery of latent concept structures in PLMs at the dataset level boils down to learning the parameters $\{\boldsymbol{\mu}_i, \boldsymbol{\Sigma}_i\}_{i=1}^K$ for the $K$ concepts. Intuitively the global parameters $\{\boldsymbol{\mu}_i, \boldsymbol{\Sigma}_i\}_{i=1}^K$ are shared across different documents, and they define a mixture of $K$ Gaussian distributions. Each Gaussian distribution describes a 'cluster' of words and their contextual word embeddings.

Similarly interpretations of PLM predictions at the data-instance level is equivalent to inferring the latent variables, i.e., document-level concept distribution vectors $\boldsymbol{\theta}_m$ and word-level concept indices $z_{mj}$. Below we highlight several important aspects of our CLDA designs.

**Attention Weights as Continuous Word Counts.** Different from typical topic models Blei et al. (2003); Blei (2012) and word embeddings Mikolov et al. (2013) that can only handle *discrete* word counts, our CLDA can handle *continuous* (virtual) word counts; this better aligns with continuous attention weights in PLMs. Specifically, we denote as $w_{mj}$ the *continuous word count* for the $j$'th word in document $m$. We explore three schemes of computing $w_{mj}$:

- **Identical Weights:** Use identical weights for different words, i.e., $w_{mj} = 1, \forall m, j$. This is equivalent to typical discrete word counts.
- **Attention-Based Weights with Fixed Length:** Use $w_{mj} = L' a_{mj}$, where $L'$ is a fixed sequence length shared across all documents.
- **Attention-Based Weights with Variable Length:** Use $w_{mj} = L_m a_{mj} / \sum_{k=1}^{L_m} a_{mk}$, where $L_m$ is true sequence length without padding. Note that in practice, $\sum_{i=1}^{L_m} a_{mk} \neq 1$ due to padding tokens in PLMs.

**Theoretical Analysis Comparing Different Schemes.** We provide theoretical analysis in the Appendix showing the advantages of attention-based schemes against the identical scheme.

**Differential Word Counts as Variables.** As discussed above, CLDA use attention weights $[a_{mj}]_{j=1}^{L_m}$ to compute virtual word counts $[w_{mj}]_{j=1}^{L_m}$ in our contextual topic models, making them continuous and differentiable. Therefore, the gradients of CLDA's learning objectives (as discussed in Sec. 3.2) w.r.t. the attention weights can backprop through the large neural PLM during joint training. This

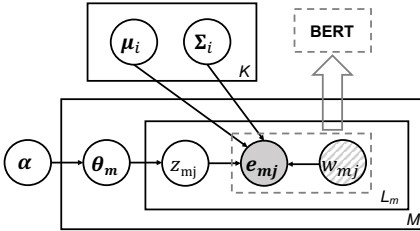

Figure 2: Graphical model of our CLDA. The *striped* circle represents *continuous* word counts.

makes it possible for CLDA to work as a PLM regulator (or regularizer) to further improving the PLM's performance via such joint training. Alternatively, we could also choose not to backprop the CLDA's learning objective through the PLM; this way, CLDA can work as a PLM interpreter without affecting the PLM's accuracy (more details in Sec. 3.3).

**Contextual Continuous Word Representations.** Note that different from topic models Blei et al. (2003) and typical word embeddings Mikolov et al. (2013); Dieng et al. (2020) where word representations are *static*, word representations in PLMs are *contextual*; specifically, the same word can have different embeddings in different documents (contexts). For example, the word 'soft' can appear as the $j_1$'th word in document $m_1$ and as the $j_2$'th word in document $m_2$, and therefore have two different embeddings (i.e., $\mathbf{e}_{m_1 j_1} \neq \mathbf{e}_{m_2 j_2}$).

Correspondingly, in our CLDA, we do not constrain the same word to have a static embedding; instead we assume that a word embedding is drawn from a Gaussian distribution corresponding to its latent topic. It is also worth noting that word representations in CLDA is continuous, which is different from typical topic models Blei et al. (2003) based on (discrete) bag-of-words representations.

## 3.2 Inference and Learning

Below we discuss the inference and learning procedure for CLDA. We start by introducing the *inference* of document-level and word-level concepts (i.e., $z_{mj}$ and $\boldsymbol{\theta}_m$) given the global concept parameters (i.e., $\{(\boldsymbol{\mu}_i, \boldsymbol{\Sigma}_i)\}_{i=1}^K$), and then introduce the *learning* of these global concept parameters.

### 3.2.1 Inference

**Inferring Document-Level and Word-Level Concepts.** We formulate the problem of interpreting PLM predictions at the concept level as inferring document-level and word-level concepts. Specifically, given global concept parameters $\{(\boldsymbol{\mu}_i, \boldsymbol{\Sigma}_i)\}_{i=1}^K$, the *contextual* word embeddings $\mathbf{e}_m \triangleq [\mathbf{e}_{mj}]_{j=1}^{L_m}$, and the associated attention weights $\mathbf{a}_m \triangleq [a_{mj}]_{j=1}^{L_m}$, a PLM produces for each document $m$, our CLDA infers the posterior distribution of the document-level concept vector $\boldsymbol{\theta}_m$, i.e., $p(\boldsymbol{\theta}_m | \mathbf{e}_m, \mathbf{a}_m, \{(\boldsymbol{\mu}_i, \boldsymbol{\Sigma}_i)\}_{i=1}^K)$, and the posterior distribution of the word-level concept index $z_{mj}$, i.e., $p(z_{mj} | \mathbf{e}_m, \mathbf{a}_m, \{(\boldsymbol{\mu}_i, \boldsymbol{\Sigma}_i)\}_{i=1}^K)$.

**Variational Distributions.** These posterior distributions are intractable; we therefore resort to variational inference Jordan et al. (1998); Blei et al. (2003) and use variational distributions $q(\boldsymbol{\theta}_m | \boldsymbol{\gamma}_m)$

and $q(z_{mj}|\boldsymbol{\phi}_{mj})$ to approximate them. Here $\boldsymbol{\gamma}_m \in \mathbb{R}^K$ and $\boldsymbol{\phi}_{mj} \triangleq [\phi_{mji}]_{i=1}^K \in \mathbb{R}^K$ are variational parameters to be estimated during inference. This leads to the following joint variational distribution:

$$q(\boldsymbol{\theta}_m, \{\mathbf{z}_{mj}\}_{j=1}^{L_m}|\boldsymbol{\gamma}_m, \{\boldsymbol{\phi}_{mj}\}_{j=1}^{L_m}) = q(\boldsymbol{\theta}_m|\boldsymbol{\gamma}_m) \cdot \prod_{j=1}^{L_m} q(z_{mj}|\boldsymbol{\phi}_{mj}) \tag{1}$$

**Evidence Lower Bound.** For each document $m$, finding the optimal variational distributions is then equivalent to maximizing the following evidence lower bound (ELBO):

$$\mathcal{L}(\boldsymbol{\gamma}_m, \{\boldsymbol{\phi}_{mj}\}_{j=1}^{L_m}; \boldsymbol{\alpha}, \{(\boldsymbol{\mu}_i, \boldsymbol{\Sigma}_i)\}_{i=1}^K) = \mathbb{E}_q[\log p(\boldsymbol{\theta}_m|\boldsymbol{\alpha})] + \sum_{j=1}^{L_m} \mathbb{E}_q[\log p(z_{mj}|\boldsymbol{\theta}_m)]$$

$$+ \sum_{j=1}^{L_m} \mathbb{E}_q[\log p(\mathbf{e}_{mj}|z_{mj}, \boldsymbol{\mu}_{z_{mj}}, \boldsymbol{\Sigma}_{z_{mj}})] - \mathbb{E}_q[\log q(\boldsymbol{\theta}_m)] - \sum_{j=1}^{L_m} \mathbb{E}_q[\log q(z_{mj})], \tag{2}$$

where the expectation is taken over the joint variational distribution in Eqn. 1.

**Likelihood with Continuous Word Counts.** One key difference between CLDA and typical topic models Blei et al. (2003); Blei (2012) is the virtual continuous word counts (discussed in Sec. 3.1). Specifically, we define the likelihood in the third term of Eqn. 2 as:

$$p(\mathbf{e}_{mj}|z_{mj}, \boldsymbol{\mu}_{z_{mj}}, \boldsymbol{\Sigma}_{z_{mj}}) = [\mathcal{N}(\mathbf{e}_{mj}; \boldsymbol{\mu}_{mj}, \boldsymbol{\Sigma}_{mj})]^{w_{mj}}. \tag{3}$$

Note that Eqn. 3 is the likelihood of $w_{mj}$ (virtual) words, where $w_{mj}$ can be a continuous value derived from the PLM's attention weights (details in Sec. 3.1).

Correspondingly, in the third item of Eqn. 2, we have:

$$\mathbb{E}_q[\log p(\mathbf{e}_{mj}|z_{mj}, \boldsymbol{\mu}_{z_{mj}}, \boldsymbol{\Sigma}_{z_{mj}})] = \sum_{m,j,i} \phi_{mji} w_{mj} \log \mathcal{N}(\mathbf{e}_{mj}|\boldsymbol{\mu}_i, \boldsymbol{\Sigma}_i)$$

$$= \sum_{m,j,i} \phi_{mji} w_{mj} \{-\tfrac{1}{2}(\mathbf{e}_{mj} - \boldsymbol{\mu}_i)^T \boldsymbol{\Sigma}_i^{-1}(\mathbf{e}_{mj} - \boldsymbol{\mu}_i) - \log[(2\pi)^{d/2}|\boldsymbol{\Sigma}_i|^{1/2}]\} \tag{4}$$

**Update Rules.** Taking the derivative of the ELBO in Eqn. 2 w.r.t. $\phi_{mji}$ (see the Appendix for details) and setting it to 0 yields the update rule for $\phi_{mji}$:

$$\phi_{mji} \propto \frac{w_{mj}}{|\boldsymbol{\Sigma}_i|^{1/2}} \exp[\Psi(\gamma_{mi}) - \Psi(\sum_{i'} \gamma_{mi'}) - \tfrac{1}{2}(\mathbf{e}_{mj} - \boldsymbol{\mu}_i)^T \boldsymbol{\Sigma}_i^{-1}(\mathbf{e}_{mj} - \boldsymbol{\mu}_i)] \tag{5}$$

with the normalization constraint $\sum_{i=1}^K \phi_{mji} = 1$.

$$\gamma_{mi} = \alpha_i + \sum_j \phi_{mji} w_{mj}, \tag{6}$$

where $\boldsymbol{\alpha} \triangleq [\alpha_i]_{i=1}^K$ is the hyperparameter for the Dirichlet prior distribution of $\boldsymbol{\theta}_m$. In summary, the inference algorithm will alternate between updating $\phi_{mji}$ for all $(m, j, i)$ tuples and updating $\boldsymbol{\gamma}_{mi}$ for all $(m, i)$ tuples.

### 3.2.2 LEARNING

**Learning Dataset-Level Concept Parameters.** The inference algorithm in Sec. 3.2.1 assumes availability of the dataset-level (global) concept parameters $\{(\boldsymbol{\mu}_i, \boldsymbol{\Sigma}_i)\}_{i=1}^K$. To learn such these parameters, one needs to iterate between (1) inferring document-level variational parameters $\boldsymbol{\gamma}_m$ as well as word-level variational parameters $\boldsymbol{\phi}_{mj}$ in Sec. 3.2.1 and (2) learning dataset-level concept parameters $\{(\boldsymbol{\mu}_i, \boldsymbol{\Sigma}_i)\}_{i=1}^K$.

**Update Rules.** Similar to Sec. 3.2.1, we expand the ELBO in Eqn. 2 (see the Appendix for details), take its derivative w.r.t. $\boldsymbol{\mu}_i$, set it to 0, yielding the update rule for learning $\boldsymbol{\mu}_i$:

$$\boldsymbol{\mu}_i = \frac{\sum_{m,j} \phi_{mji} w_{mj} \mathbf{e}_{mj}}{\sum_{m,j} \phi_{mji} w_{mj}}, \tag{7}$$

Similarly, setting the derivatives w.r.t. $\boldsymbol{\Sigma}$ to 0, we have

$$\boldsymbol{\Sigma}_i = \frac{\sum_{m,j} \phi_{mji} w_{mj} (\mathbf{e}_{mj} - \boldsymbol{\mu}_i)(\mathbf{e}_{mj} - \boldsymbol{\mu}_i)^T}{\sum_{m,j} \phi_{mji} w_{mj}}. \tag{8}$$

**Effect of Attention Weights.** From Eqn. 7 and Eqn. 8, we can observe that the attention weight of the $j$'th word in document $m$, i.e., $a_{mj}$, affects the virtual continuous word count $w_{mj}$ (see Sec. 3.1), thereby affecting the update of the dataset-level concept center and covariance $\boldsymbol{\mu}_i$ and $\boldsymbol{\Sigma}_i$.

Specifically, if we use attention-based weights with fixed length or variable length in Sec. 3.1, the continuous word count $w_{mj}$ will be proportional to the attention weight $a_{mj}$. Therefore, when updating the concept center $\boldsymbol{\mu}_i$ as a weighted average of different word embeddings $\mathbf{e}_{mj}$, CLDA naturally places more focus on words with higher attention weights $a_{mj}$ from PLMs, consequently making the interpretations sharper and improving performance (see Sec. 4.2 for detailed results and the Appendix for theoretical analysis).

Interestingly, we also observe that PLMs' attention weights on stop words such as 'the' and 'a' tend to be much lower; therefore CLDA can naturally ignore these

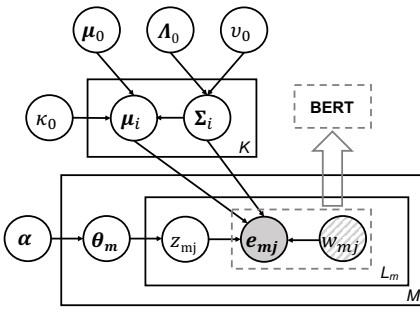

Figure 3: Probabilistic graphical model of our smoothed CLDA.

concept-irrelevant stop words when learning and inferring concepts (topics). This is in contrast to typical topic models Blei et al. (2003); Blei (2012) that require preprocessing to remove stop words.

**Smoothing with Prior Distributions on $\{(\boldsymbol{\mu}_i, \boldsymbol{\Sigma}_i)\}_{i=1}^K$.** To alleviate overfitting and prevent singularity in numerical computation, we impose priors distributions on $\boldsymbol{\mu}_i$ and $\boldsymbol{\Sigma}_i$ to smooth the learning process (Fig. 3). Specifcally, we use a Normal-Inverse-Wishart prior on $\boldsymbol{\mu}_i$ and $\boldsymbol{\Sigma}_i$ as follows:

$$\boldsymbol{\Sigma}_i \sim \mathcal{IW}(\boldsymbol{\Lambda}_0, \nu_0), \quad \boldsymbol{\mu}_i | \boldsymbol{\Sigma}_i \sim \mathcal{N}(\boldsymbol{\mu}_0, \boldsymbol{\Sigma}_k / \kappa_0),$$

where $\boldsymbol{\Lambda}_0$, $\nu_0$, $\boldsymbol{\mu}_0$, and $\kappa_0$ are hyperparameters for the prior distributions. With the prior distribution above, the update rules for $\boldsymbol{\mu}_i$ and $\boldsymbol{\Sigma}_i$ become:

$$\boldsymbol{\mu}_i = \frac{k_0 \boldsymbol{\mu}_0 + n \widetilde{\boldsymbol{\mu}}_i}{k_0 + n}, \quad \boldsymbol{\Sigma}_i = \frac{\boldsymbol{\Lambda}_0 + \mathbf{S}_i + \frac{\kappa_0 n}{k_0 + n} (\widetilde{\boldsymbol{\mu}}_i - \boldsymbol{\mu}_0)(\widetilde{\boldsymbol{\mu}}_i - \boldsymbol{\mu}_0)^T}{\nu_0 + n - K - 1}, \tag{9}$$

$$\mathbf{S}_i = \sum\nolimits_{m,j} \phi_{mji} w_{mj} (\mathbf{e}_{mj} - \widetilde{\boldsymbol{\mu}}_i)(\mathbf{e}_{mj} - \widetilde{\boldsymbol{\mu}}_i)^T, \tag{10}$$

where $n = \sum_{m,j} \phi_{mji} w_{mj}$ is the total virtual word counts used to estimate $\boldsymbol{\mu}_i$ and $\boldsymbol{\Sigma}_i$. Eqn. 9 is the smoothed version of Eqn. 7 and Eqn. 8, respectively. From the Bayesian perfective, they correspond to the expectations of $\boldsymbol{\mu}_i$'s and $\boldsymbol{\Sigma}_i$'s posterior distributions (see the Appendix for detailed derivation).

**Online Learning of $\boldsymbol{\mu}_i$ and $\boldsymbol{\Sigma}_i$.** PLMs are deep neural networks trained using minibatches of data, while Eqn. 7 and Eqn. 8 need to go through the whole dataset before each update. We therefore use exponential moving average (EMA) to work with minibatchs (see the Appendix for details).

### 3.3 CLDA as PLM Interpreters and Regulators

CLDA can be used as either a PLM interpreter, which interprets model predictions at the concept level without any performance sacrifice, or a PLM regulator, which is jointly trained with PLMs during finetuning to further improve performance. This is possible because both the word counts and contextual word embeddings in CLDA is *continuous* and differentiable. Below we start with more details on the differentiability and then introduce our CLDA interpreter and CLDA regulator.

---

**Algorithm 1** Algorithm for CLDA Regulators (w/o EMA)

---

**Input:** Pretrained $f_{ae}(\cdot)$ and $f_c(\cdot)$, initialized $g(\cdot)$, initialized $\{\boldsymbol{\gamma}_m\}_{m=1}^M$, $\{\boldsymbol{\Phi}\}_{m=1}^M$, and $\{\boldsymbol{\Omega}\}_{m=1}^M$, documents $\{\mathcal{D}\}_{m=1}^M$, number of epochs T.
**for** $t = 1 : T$ **do**
    **for** $m = 1 : M$ **do**
        Update $\boldsymbol{\Phi}_m$ and $\boldsymbol{\gamma}_m$ using Eqn. 5 and Eqn. 6, respectively.
        Update $f_{ae}(\cdot)$, $f_c(\cdot)$, and $g(\cdot)$ using Eqn. 11.
    Update $\boldsymbol{\Omega}$ using Eqn. 9.

---

**Differentiable Continuous Word Counts and Contextual Word Embeddings.** One of CLDA's advantage is that it handle *continuous* word counts and word embeddings. Such continuity translates to better differentability, and therefore is particularly desirable when ones wants to jointly train CLDA and a PLM to further improve PLM performance.

As shown in Fig. 2, CLDA connects to a PLM through the attention weights $a_{mj}$ (related to the word counts $w_{mj}$) and contextual word embeddings $\mathbf{e}_{mj}$. Therefore, if the CLDA learning objective

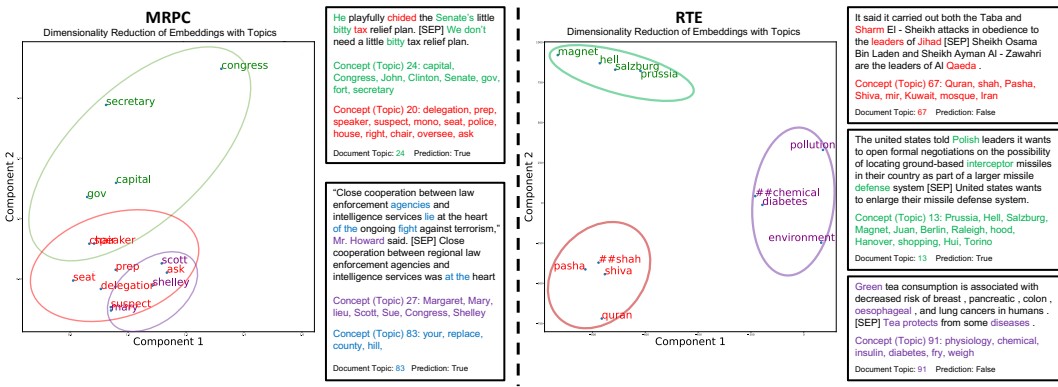

Figure 4: Visualization of CLDA's learned topics of contextual word embeddings. **Left:** MRPC's dataset-level interpretation with two example documents. Concept 83 is relatively far from the other three concepts in the embedding space; therefore we omit it on the left panel for better readability. **Right:** RTE's dataset-level interpretation with three example documents.

(Eqn. 2) differentiable w.r.t. $a_{mj}$ and $\mathbf{e}_{mj}$, these gradients can then be propagated to the PLM parameters and help finetune the PLM. The derivative of the ELBO in Eqn. 2 w.r.t. $\mathbf{e}_{mj}$ is: $\frac{\partial L}{\partial \mathbf{e}_{mj}} = \sum_i \phi_{mji} w_{mj} \boldsymbol{\Sigma}_i^{-1}(\boldsymbol{\mu}_i - \mathbf{e}_{mj})$. Similarly we can get the derivative w.r.t. $a_{mj}$ using the chain rule, where $\frac{\partial w_{mj}}{\partial a_{mj}}$ depends on the choice of schemes for computing $w_{mj}$ from $a_{mj}$ (described inSec. 3.1), and $\frac{\partial L}{\partial w_{mj}}$ is: $\frac{\partial L}{\partial w_{mj}} = \sum_i \phi_{mji}(\boldsymbol{\mu}_i - \mathbf{e}_{mj})^T \boldsymbol{\Sigma}_i^{-1}(\boldsymbol{\mu}_i - \mathbf{e}_{mj})$.

**CLDA as a PLM Interpreter.** Using CLDA as a PLM interpreter is straightforward. One only needs to first learn the global concept parameters $\boldsymbol{\mu}_i$ and $\boldsymbol{\Sigma}_i$ according to Sec. 3.2.2, and then infer document-level concept vectors $\boldsymbol{\theta}_m$ and word-level concept indices $\mathbf{z}_{mj}$. Together, they provide dataset-level, document-level, and word-level conceptual interpretations for PLM predictions.

**CLDA as a PLM Regulator.** One could also use CLDA as a regulator (or regularizer) when finetuning a PLM. Assume a PLM that produces attention weights and contextual word embeddings for document $m$, i.e., $(\mathbf{a}_m, \mathbf{e}_m) = f_{ae}(\mathcal{D}_m)$, as well as the CLS embedding $\mathbf{c}_m = f_c(\widetilde{\mathbf{c}}_m, \mathbf{a}_m, \mathbf{e}_m)$; here $\widetilde{\mathbf{c}}_m$ is the CLS embedding of the second-last layer. To better see the connection between CLDA and PLMs, we can rewrite the ELBO in Eqn. 2 as:

$$L_c\Big(\boldsymbol{\gamma}_m, \boldsymbol{\Phi}_m; \boldsymbol{\Omega}, \mathbf{a}_m, \mathbf{e}_m\Big) = L_c\Big(\boldsymbol{\gamma}_m, \boldsymbol{\Phi}_m; \boldsymbol{\Omega}, f_{ae}(\mathcal{D}_m)\Big),$$

where $\boldsymbol{\Phi}_m \triangleq \{\boldsymbol{\phi}_{mj}\}_{j=1}^{L_m}$ is the collection of word-level concept parameters for document $m$, and $\boldsymbol{\Omega} = \{(\boldsymbol{\mu}_i, \boldsymbol{\Sigma}_i)\}_{i=1}^K$ is the global concept parameters. Assuming a document-level predictor $g(\mathbf{c}_m)$ and denoting the ground-truth label as $y_m$, we have the PLM loss during finetuning:

$$L_p\Big(g(f_c(\widetilde{\mathbf{c}}_m, f_{ae}(\mathcal{D}_m))), y_m\Big).$$

Putting them together, we have the joint loss:

$$L_j = L_p\Big(g(f_c(\widetilde{\mathbf{c}}_m, f_{ae}(\mathcal{D}_m))), y_m\Big) + \lambda L_c\Big(\boldsymbol{\gamma}_m, \boldsymbol{\Phi}_m; \boldsymbol{\Omega}, f_{ae}(\mathcal{D}_m)\Big). \quad (11)$$

When finetuning a PLM, one can iterate between updating (1) the PLM parameters in $f_{ae}(\cdot)$, $f_c(\cdot)$, and $g(\cdot)$, (2) the CLDA global concept parameters $\boldsymbol{\Omega}$, and (3) the CLDA document-level and word-level concept parameters $\boldsymbol{\gamma}_m$ and $\boldsymbol{\Phi}_m$. Note that $f_{ae}(\cdot)$ contains most of the parameters in PLMs and appears in the CLDA loss term $L_c$; therefore CLDA can improve the finetuning process in PLMs. Alg. 1 shows an overview of CLDA training when used as a PLM regulator (to prevent clutter, we show the version without EMA).

**Theoretical Analysis.** In the Appendix, we provide theoretical guarantees that under mild assumptions our CLDA can learn concept-level interpretations for PLMs, especially in noisy data.

## 4 EXPERIMENTS

### 4.1 EXPERIMENT SETUP

**Datasets.** We use the GLUE benchmark Wang et al. (2018) to evaluate our methods. This benchmark includes multiple sub-tasks of predictions, with the paired sentences as inputs. In this paper, we use six datasets from GLUE (CoLA, MRPC, STS-B, QQP, RTE, and SST-2) to perform evaluation.

**Implementation.** Our training/validation/test data split of GLUE datasets follows exactly Devlin et al. (2018). All PLMs are base models from vanilla settings, with the hidden dimension of 768. The BERT Models are optimized by AdamW Kingma & Ba (2014) Optimizer, using a learning rate of $10^{-4}$ with linear warmup and linear learning rate decay. We finetune the models until metrics on validation sets get the highest score (see the Appendix for more implementation details).

**Evaluation Metrics.** We follow previous work Devlin et al. (2018) on the GLUE benchmark, and use the provided testing scores as our evaluation metrics for the 4 datasets, i.e., Matthew's correlation for CoLA, F1 score and accuracy for MRPC and QQP, Pearson/Spearman correlation for STS-B, and accuracy for RTE and SST-2.

**Baselines**. We compare our methods with the following four baselines: **BERT** Devlin et al. (2018), **BART** Lewis et al. (2019), **RoBERTa** Liu et al. (2019), and **DeBERTa** He et al. (2021). The basic settings of CLDA-augmented PLMs at the BERT side are kept consistent with the Base models for fair comparison (see the Appendix for details).

### 4.2 CONCEPTUAL INTERPRETATION (MORE RESULTS IN THE APPENDIX)

**Dataset-Level Interpretations.** To showcase CLDA's capability as a PLM interpreter, we use CLDA trained on MRPC and RTE, respectively, sample 3 concepts (topics) for each dataset, and plot the word embeddings of the top words (closest to the center $\boldsymbol{\mu}_i$) in these concepts using PCA. Fig. 4(left) shows the concepts from MRPC. We can observe Concept 20 is mostly about policing, including words such as 'suspect', 'police', and 'house'. Concept 24 is mostly about politics, including words such as 'capital', 'Congress', and 'Senate'. Concept 27 contains mostly names such as 'Margaret' and 'Mary'. Similarly, Fig. 4(right) shows the concepts from RTE. We can observe Concept 67 is related to Islam and includes words such as 'Quran' and 'Pasha'. Concept 13 is related to Europe and includes European countries/names such as 'Prussia' and 'Salzburg'. Concept 91 is healthcare-related and includes words such as 'physiology' and 'insulin'.

**Document-Level Interpretations.** For document-level conceptual interpretations, we sample two example documents from MRPC (Fig. 4(left)) and three from RTE (Fig. 4(right)), where each document contains a pair of sentences. The MRPC task is to predict whether one sentence paraphrases the other. For example, in the first document of MRPC, we can see that our CLDA correctly interprets the model prediction 'True' with Concept 24 (politics). The RTE task is to predict whether one sentence entail the other. For example, in the second document of RTE, CLDA correctly interprets the model prediction 'True' with Concept 13 (countries).

**Word-Level Interpretations.** For word-level conceptual interpretations, we can observe that CLDA interpret the PLM's prediction on MRPC's first document (Fig. 4(left)) using words such as 'senate' and 'bitty' that are related to politics. Note that the word 'bitty' is commonly used (with 'little') by politicians to refer to the small size of tax relief/cut plans. Similarly, for RTE's first document (Fig. 4(right)), CLDA correctly identifies Concept 67 (Islam) and interprets the model prediction 'False' by distinguishing between keywords such as 'Jihad' and 'Al Qaeda'.

### 4.3 QUANTITATIVE RESULTS

To evaluate CLDA as a PLM regulator, we use BERT, RoBERTa, BART, and DeBERTa as base models for our CLDA, leading four different CLDA models, BERT-CLDA, RoBERTa-CLDA, BART-CLDA, and DeBERTa-CLDA, respectively. Table 1 shows the performance of our CLDA variants and the correspondingly base PLMs on six benchmark datasets, CoLA, MRPC, STS-B, QQP, RTE, and SST-2.

Table 1: Results on GLUE benchmark datasets.

| Dataset
Metrics | CoLA
Matthew's Corr. | MRPC
F1/Acc. | STS-B
Pear./Spea. Corr. | QQP
F1/Acc. | RTE
Acc. | SST-2
Acc. |
|---|---|---|---|---|---|---|
| BERT-Base | 51.2 | 73.3/66.6 | 82.5/81.2 | 66.0/85.1 | 57.1 | 88.2 |
| BERT-CLDA (Ours) | **52.2** | **82.2/74.6** | **84.0/82.8** | **70.3/88.7** | **60.3** | **92.2** |
| RoBERTa-Base | 59.0 | **79.9**/66.5 | 83.3/82.2 | 52.5/78.1 | 67.4 | 93.8 |
| RoBERTa-CLDA (Ours) | **59.1** | 78.2/**71.4** | **83.5/82.4** | **69.5/88.1** | **70.1** | **94.7** |
| BART-Base | 46.9 | 85.4/79.8 | **87.6/86.5** | 70.5/88.8 | 62.6 | 91.9 |
| BART-CLDA (Ours) | **49.5** | **85.6/80.0** | **87.6/86.6** | **70.7/88.9** | **64.7** | **92.6** |
| DeBERTa-Base | 57.2 | 86.3/81.8 | 89.6/88.8 | 70.4/88.9 | 64.8 | 92.4 |
| DeBERTa-CLDA (Ours) | **59.8** | **86.7/82.1** | **89.8/89.0** | **71.5/89.1** | **68.4** | **94.0** |
| Average-Base | 53.7 | 81.2/73.7 | 85.8/84.7 | 64.9/85.2 | 63.0 | 91.6 |
| Average-CLDA (Ours) | **54.9** | **83.2/77.0** | **86.2/85.2** | **70.5/88.7** | **65.9** | **93.4** |

The last two rows show the average predictive performance across different PLM base models for different datasests. We can observe that on average, our CLDA significantly outperforms the baselines in all datasets. Notably, in the largest dataset, QQP, our CLDA improves upon the baselines by $5.6\%$ in terms of F1 score and by $3.5\%$ in terms of accuracy. Moreover, even for 'difficult' natural language inference tasks such as RTE, CLDA can still improve the average accuracy by a large margin of $2.9\%$. When using RoBERTa as the base model, CLDA achieves absolute improvements of $17.0\%$ and $10.0\%$ for F1 score and accuracy, respectively. Note that STS-B and SST-2 are relatively 'easy' datasets, and even the BERT-Base model could achieve correlation higher than $82\%$ and accuracy higher than $88\%$, respectively. In this case the room for improvement is minimal. However, our CLDA could still lead to slight improvement in terms of Pearson correlation and Spearman correlation for STS-B, as well as reasonable accuracy improvement for SST-2.

## 4.4 ABLATION STUDIES

To evaluate different schemes of computing the virtual word counts $w_{mj}$ from attention weights (as introduced in Sec. 3.1), we perform ablation studies on the CoLA dataset using different PLM base models. Table 2 shows the results on the base model and CLDA with identical weights (CLDA-Identical), attention-based weights with variable length (CLDA-Variable), and attention-based weights with fixed length (CLDA-Fixed).

Table 2: Ablation studies for the CoLA dataset in terms of Matthew's correlation.

| Model | BERT | RoBERTa | BART | DeBERTa |
|---|---|---|---|---|
| Base | 51.2 | 59.0 | 46.9 | 57.2 |
| CLDA-Identical | 50.4 | 55.6 | 47.0 | 51.9 |
| CLDA-Variable | **52.2** | 57.5 | 47.4 | 59.5 |
| CLDA-Fixed | **52.2** | **59.1** | **49.5** | **59.8** |

One observation is that CLDA-Identical tends to underperform the PLM base models, while both CLDA-Variable and CLDA-Fixed can significantly outperform the base models. This verifies the importance of using attention weights to compute the virtual continuous word counts. Another interesting observation is that CLDA-Fixed slightly outperforms CLDA-Variable. Note that both CLDA-Fixed and CLDA-Variable use attention weights to compute virtual continuous word counts; the difference is that in CLDA-Variable assigns longer documents more (total) weights when learning the global concept parameters (Eqn. 7 and Eqn. 8), while CLDA-Fixed treats each document fairly (as if they had fixed length). Therefore Table 2 shows that it is beneficial to assume different documents have fixed total virtual word counts for CLDA.

## 5 CONCLUSION

We develop CLDA as a genearal framework to interpret pretrained word embeddings at the concept level. Our CLDA is compatible with any attention-based PLMs. It can not only interpret how PLMs make predictions, but also help improve contextual word embeddings in an end-to-end manner, thereby boosting predictive performance.

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

## A  DETAILS ON LEARNING CLDA

**Update Rules.** Similar to Sec. 3.2.1 of the main paper, we expand the ELBO in Eqn. 2 of the main paper, take its derivative w.r.t. $\boldsymbol{\mu}_i$ and set it to 0: $\frac{\partial L}{\partial \boldsymbol{\mu}_i} = \sum_{m,j} \phi_{mji} w_{mj} \boldsymbol{\Sigma}_i^{-1} (\mathbf{e}_{mj} - \boldsymbol{\mu}_i) = 0$, yielding the update rule for learning $\boldsymbol{\mu}_i$:

$$\boldsymbol{\mu}_i = \frac{\sum_{m,j} \phi_{mji} w_{mj} \mathbf{e}_{mj}}{\sum_{m,j} \phi_{mji} w_{mj}}, \tag{12}$$

where $\boldsymbol{\Sigma}_i^{-1}$ is canceled out. Similarly, setting the derivatives w.r.t. $\boldsymbol{\Sigma}$ to 0, i.e., $\frac{\partial L}{\partial \boldsymbol{\Sigma}_i} = \frac{1}{2} \sum_{m,j} \phi_{mji} w_{mj} (-\boldsymbol{\Sigma}_i^{-1} + \boldsymbol{\Sigma}_i^{-1} (\mathbf{e}_{mj} - \boldsymbol{\mu}_i)(\mathbf{e}_{mj} - \boldsymbol{\mu}_i)^T \boldsymbol{\Sigma}_i^{-1})$, we have

$$\boldsymbol{\Sigma}_i = \frac{\sum_{m,j} \phi_{mji} w_{mj} (\mathbf{e}_{mj} - \boldsymbol{\mu}_i)(\mathbf{e}_{mj} - \boldsymbol{\mu}_i)^T}{\sum_{m,j} \phi_{mji} w_{mj}}. \tag{13}$$

**Online Learning of $\boldsymbol{\mu}_i$ and $\boldsymbol{\Sigma}_i$.** Note that PLMs are deep neural networks trained using minibatches of data, while Eqn. 7 and Eqn. 8 need to go through the whole dataset before each update. Inspired by Hoffman et al. (2010); Oord et al. (2017), we using exponential moving average (EMA) to work with minibatchs. Specifically, we update them as:

$$\boldsymbol{\mu}_i \leftarrow \rho \cdot N \cdot \boldsymbol{\mu}_i + (1 - \rho) \cdot B \cdot \widetilde{\boldsymbol{\mu}}_i,$$
$$\boldsymbol{\Sigma}_i \leftarrow \rho \cdot N \cdot \boldsymbol{\Sigma}_i + (1 - \rho) \cdot B \cdot \widetilde{\boldsymbol{\Sigma}}_i,$$
$$N \leftarrow \rho \cdot N + (1 - \rho) \cdot B,$$
$$\boldsymbol{\mu}_i \leftarrow \frac{\boldsymbol{\mu}_i}{N}, \quad \boldsymbol{\Sigma}_i \leftarrow \frac{\boldsymbol{\Sigma}_i}{N},$$

where $B$ is the minibatch size, $N$ is a running count, and $\rho \in (0, 1)$ is the momentum hyperparameter. $\widetilde{\boldsymbol{\mu}}_i$ and $\widetilde{\boldsymbol{\Sigma}}_i$ are the updated $\boldsymbol{\mu}_i$ and $\boldsymbol{\Sigma}_i$ after applying Eqn. 7 and Eqn. 8 only on the *current minibatch*.

Table 3: Example concepts on RTE dataset learned by CLDA.

| Concepts | Top Words | | | | | | | |
|---|---|---|---|---|---|---|---|---|
| **bio-chem** | cigarette | biological | ozone | cardiovascular | chemist | liver | chemical | toxin |
| **citizenship** | indies | bolivian | fiji | surrey | jamaican | dutch | latino | caribbean |
| **names** | mozart | spielberg | einstein | bush | kurt | liszt | hilton | lynn |
| **conspiracy** | secretly | corrupt | disperse | infected | ill | hidden | illegally | sniper |
| **administration** | reagan | interior | ambassador | prosecutor | diplomat | legislative | spokesman | embassy |
| **crime** | fraud | laundering | sheriff | prosecutor | corruption | fool | robber | greed |

**Paired Sentences as a Document.** Many modern natural language processing tasks involve predicting a label from a pair of sentences (for example, given two sentences, predict whether one sentence paraphrases the other). In this case, one document may contain a pair of sentences (with length $L_{m_1}$ and $L_{m_2}$, and $L_m = L_{m_1} + L_{m_2}$) as PLM inputs, and $\gamma$ of each sentence can be inferred as: $\gamma_{m_1 i} = \alpha_i + \sum_{j=0}^{L_{m_1}} \phi_{mji} w_{mj}$, $\gamma_{m_2 i} = \alpha_i + \sum_{j=m_1}^{L_m} \phi_{mji} w_{mj}$.

## B  EXPERIMENTAL SETTINGS AND IMPLEMENTATION DETAILS

We will release all code, models, and data. Below we provide more details on the experimental settings and practical implementation.

**Data Preprocessing.** Our training/validation/test data split of GLUE datasets follows exactly Devlin et al. (2018). We train our model on the training data, perform model selection (select hyperparameters) using the validation data, and evaluate methods on the test data. Our tokenization and bert-configurations follow previous PLMs that we compare with. For fair comparison, we use lowercase tokenization both in base models and our CLDA models. According to different versions of CLDA weighting, we can choose whether to calculate TF-IDF scores in documents. If we use an identical-weight CLDA, additional computing of TF-IDF scores is necessary to filter words with little information for our CLDA-based topic models.

**Implementation.** All PLMs are base models from vanilla settings, with the hidden dimension of 768. We initialize the models with the seed 2021. The BERT Models are optimized by AdamW Kingma & Ba (2014) Optimizer, using a learning rate of $10^{-4}$ with linear warmup and linear learning rate decay. We finetune the models until metrics on validation sets get the highest score. We treat the training batch-size, CLDA prior parameters, and $\lambda$ (in Eqn. 11 of the main paper) as hyperparameters, and run grid-search in training and validation to search for models with the highest possible performance. To alleviate overfitting of CLDA during joint training, we periodically include the CLDA loss term $L_c$ in the joint loss $L_j$ along epochs, i.e., using the CLDA term every 1/3/5 epochs (as a training hyperparameter as well), along with original base PLM finetuning loss. We use the **fixed** scheme for CLDA training by default to produce the results in Table 1 of the main paper. We use the penultimate-layer word embeddings for CLDA in Eqn. 3 of the main paper, because our preliminary results show that using the penultimate layer instead of the output layer improves performance.

**Baselines.** To ensure fair comparison, during fine-tuning, we select the epoch for both baselines and CLDA entirely based on validation accuracy and report the test accuracy; in contrast, [Devlin et al., 2018] directly chooses the 3rd epoch; we argue that this is not rigorous and potentially 'overfits' test sets. Also, as aforementioned, We follow the convention of topic models and preprocess the documents into lower-case words for both baselines and CLDA; in contrast, [Devlin et al., 2018] keeps the words unchanged. Nevertheless, note that our CLDA can interpret any PLMs without accuracy sacrifice; therefore the exact accuracy for BERT-base is less relevant in our case.

**Visualization Postprocessing.** For better showcase the dataset-level concepts as in Fig. 4 of the main paper, we may employ simple linear transformations on the embedding of words after the aforementioned PCA step, in order to scatter all the informative words on the same figures. However, for some datasets such as STS-B, this is not necessary so we don't use it.

**Topic (Concept) Identification.** Inspired by Blei et al. (2003), we identify meaningful topics by listing the top-5 topics for each word, computing the inverse document frequency (IDF), and filtering out topics with the lowest IDF scores. Note that although GLUE benchmark are datasets that consists of documents with small size, making it particularly challenging for traditional topic

models (such as LDA) to learn topics; interestingly our CLDA can still do well in learning the topics. We contribute this to the following observations: (1) Compared to traditional LDA using *discrete* word representations, CLDA uses *continuous* word embeddings. In such a continuous space, topics learned for one word can also help neighboring words; this alleviates the sparsity issue caused by short documents and therefore learns better topics. (2) CLDA's attention-based continuous word counts further improves sample efficiency. In CLDA, important words have larger attention weights and therefore larger continuous word counts. In this case, *one* important word in a sentence possesses statistical (sample) power equivalent to *multiple* words; this leads to better sample efficiency in CLDA.

## C  EXPANSION OF ELBO

We can expand the ELBO in Eqn. 2 of the main paper as:

$$
\begin{aligned}
\mathcal{L}(\boldsymbol{\gamma}, \boldsymbol{\phi}; \boldsymbol{\alpha}, \{\boldsymbol{\mu}\}_K, \{\boldsymbol{\Sigma}\}_K) = & \log \boldsymbol{\Gamma}(\sum_{i=1}^{K} \alpha_i) - \sum_{i=1}^{K} \log \boldsymbol{\Gamma}(\alpha_i) + \sum_{i=1}^{K} (\alpha_i - 1)(\Psi(\boldsymbol{\gamma}_i) - \Psi(\sum_{j=1}^{K} \boldsymbol{\gamma}_j)) \\
& + \sum_{j=1}^{L} \sum_{i=1}^{K} \phi_{ji}(\Psi(\boldsymbol{\gamma}_i) - \Psi(\sum_{k=1}^{K} \boldsymbol{\gamma}_k)) \\
& + \sum_{m,j,i} \phi_{mji} w_{mj} \{-\tfrac{1}{2}(\mathbf{e}_{mj} - \boldsymbol{\mu}_i)^T \boldsymbol{\Sigma}_i^{-1}(\mathbf{e}_{mj} - \boldsymbol{\mu}_i) \\
& \qquad\qquad\qquad\qquad - \log[(2\pi)^{d/2}|\boldsymbol{\Sigma}_i|^{1/2}]\} \\
& - \log \boldsymbol{\Gamma}(\sum_{j=1}^{K} \boldsymbol{\gamma}_j) + \sum_{i=1}^{K} \log \boldsymbol{\Gamma}(\boldsymbol{\gamma}_i) - \sum_{i=1}^{K} (\boldsymbol{\gamma}_i - 1)(\Psi(\boldsymbol{\gamma}_i) - \Psi(\sum_{j=1}^{K} \boldsymbol{\gamma}_j)) \\
& - \sum_{j=1}^{L} \sum_{i=1}^{K} \phi_{ji} \log \phi_{ji}.
\end{aligned}
\tag{14}
$$

## D  DERIVATION ON SMOOTHED CLDA

To alleviate overfitting and prevent singularity in numerical computation, we impose priors distributions on $\boldsymbol{\mu}_i$ and $\boldsymbol{\Sigma}_i$ to smooth the learning process. Specifcally, we use a Normal-Inverse-Wishart prior on $\boldsymbol{\mu}_i$ and $\boldsymbol{\Sigma}_i$ as follows:

$$
\begin{aligned}
\boldsymbol{\Sigma}_i &\sim \mathcal{IW}(\boldsymbol{\Lambda}_0, \nu_0), \\
\boldsymbol{\mu}_i | \boldsymbol{\Sigma}_i &\sim \mathcal{N}(\boldsymbol{\mu}_0, \boldsymbol{\Sigma}_k / \kappa_0),
\end{aligned}
$$

where $\boldsymbol{\Lambda}_0$, $\nu_0$, $\boldsymbol{\mu}_0$, and $\kappa_0$ are hyperparameters for the prior distributions. With the prior distribution above, the update rules for the parameters of the posterior distribution $\mathcal{NIW}(\boldsymbol{\mu}_i, \boldsymbol{\Sigma}_i | \boldsymbol{\mu}_i^{(n)}, \boldsymbol{\Lambda}_i^{(n)}, \kappa_i^{(n)}, \nu_i^{(n)})$ become:

$$
\boldsymbol{\mu}_i^{(n)} = \frac{k_0 \boldsymbol{\mu}_0 + n \widetilde{\boldsymbol{\mu}}_i}{k_0 + n},
\tag{15}
$$

$$
\boldsymbol{\Lambda}_i^{(n)} = \boldsymbol{\Lambda}_0 + \mathbf{S}_i + \frac{\kappa_0 n}{k_0 + n}(\widetilde{\boldsymbol{\mu}}_i - \boldsymbol{\mu}_0)(\widetilde{\boldsymbol{\mu}}_i - \boldsymbol{\mu}_0)^T,
\tag{16}
$$

$$
\kappa_i^{(n)} = \kappa_0 + n,
\tag{17}
$$

$$
\nu_i^{(n)} = \nu_0 + n,
\tag{18}
$$

$$
\mathbf{S}_i = \sum_{m,j} \phi_{mji} w_{mj} (\mathbf{e}_{mj} - \widetilde{\boldsymbol{\mu}}_i)(\mathbf{e}_{mj} - \widetilde{\boldsymbol{\mu}}_i)^T,
\tag{19}
$$

where $n = \sum_{m,j} \phi_{mji} w_{mj}$ is the total virtual word counts used to estimate $\boldsymbol{\mu}_i$ and $\boldsymbol{\Sigma}_i$. Taking the expectations of $\boldsymbol{\mu}_i$ and $\boldsymbol{\Sigma}_i$ over the posterior distibution $\mathcal{NIW}(\boldsymbol{\mu}_i, \boldsymbol{\Sigma}_i | \boldsymbol{\mu}_i^{(n)}, \boldsymbol{\Lambda}_i^{(n)}, \kappa_i^{(n)}, \nu_i^{(n)})$, we

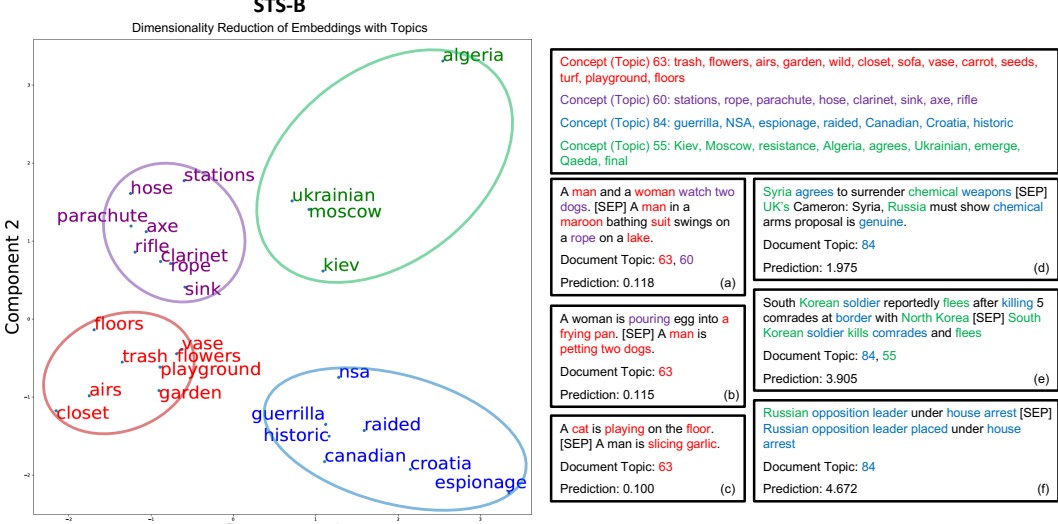

Figure 5: Visualization of CLDA's learned topics of contextual word embeddings. We show STS-B's dataset-level interpretation with six example documents. The prediction of CLDA is between the range of $[0, 5]$.

have the update rules as:

$$\boldsymbol{\mu}_i \leftarrow \mathbb{E}_{\mathcal{NIW}}[\boldsymbol{\mu}_i] = \frac{k_0 \boldsymbol{\mu}_0 + n \widetilde{\boldsymbol{\mu}}_i}{k_0 + n}, \tag{20}$$

$$\boldsymbol{\Sigma}_i \leftarrow \mathbb{E}_{\mathcal{NIW}}[\boldsymbol{\Sigma}_i] = \frac{\boldsymbol{\Lambda}_0 + \mathbf{S}_i + \frac{\kappa_0 n}{k_0 + n}(\widetilde{\boldsymbol{\mu}}_i - \boldsymbol{\mu}_0)(\widetilde{\boldsymbol{\mu}}_i - \boldsymbol{\mu}_0)^T}{\nu_0 + n - K - 1}, \tag{21}$$

$$\mathbf{S}_i = \sum_{m,j} \phi_{mji} w_{mj} (\mathbf{e}_{mj} - \widetilde{\boldsymbol{\mu}}_i)(\mathbf{e}_{mj} - \widetilde{\boldsymbol{\mu}}_i)^T. \tag{22}$$

Eqn. 20 and Eqn. 21 are the smoothed version of Eqn. 7 and Eqn. 8, respectively. From the Bayesian perspective, they correspond to the expectations of $\boldsymbol{\mu}_i$'s and $\boldsymbol{\Sigma}_i$'s posterior distributions.

## E    MORE CONCEPTUAL INTERPRETATION RESULTS

**Dataset-Level Interpretations.**    As in the main paper, we leverage CLDA as interpreter on STS-B and QQP, respectively, sample 4 concepts (topics) for each dataset, and plot the word embeddings of the top words (closest to the center $\boldsymbol{\mu}_i$) in these concepts using PCA. Fig. 5 shows the concepts from STS-B. We can observe Concept 63 is mostly about household and daily life, including words such as 'trash', 'flowers', 'airs', and 'garden'. Concept 60 is mostly about tools, including words such as 'stations', 'rope', 'parachute', and 'hose'. Concept 84 is mostly about national security, including words such as 'guerilla', 'NSA', 'espionage', and 'raided'. Concept 55 contains mostly countries and cities such as 'Kiev', 'Moscow', 'Algeria', and 'Ukrainian'. Similarly, Fig. 6 shows the concepts from QQP. We can observe Concept 12 is mostly about negative attitude, including words such as 'boring', 'criticism', and 'blame'. Concept 73 is mostly about Psychology, including words such as 'adrenaline', 'haunting', and 'paranoia'. Concept 34 is mostly about prevention and conservatives, including words such as 'destroys', 'unacceptable', and 'prohibits'. Concept 64 is mostly about strategies, including words such as 'rumours', 'boycott', and 'deportation'.

**Document-Level Interpretations.** For document-level conceptual interpretations, we sample six example documents from STS-B (Fig. 5) and eight from QQP (Fig. 6), respectively, where each document contains a pair of sentences. The STS-B task is to predict the semantic similarity between two sentences with the score range of $[0, 5]$. For example, in Document (a) of Fig. 5, we can see that CLDA correctly interpret the model's predicted similarity score '0.118' (which is relatively low,) with Concept 63 (household and daily life) and Concept 60 (tools). Similarly, in Document (f) of Fig. 5, we can see that CLDA correctly interpret the model's predicted similarity score '4.672' (which is relatively high) with Concept 84 (national security). The QQP task is to predict whether

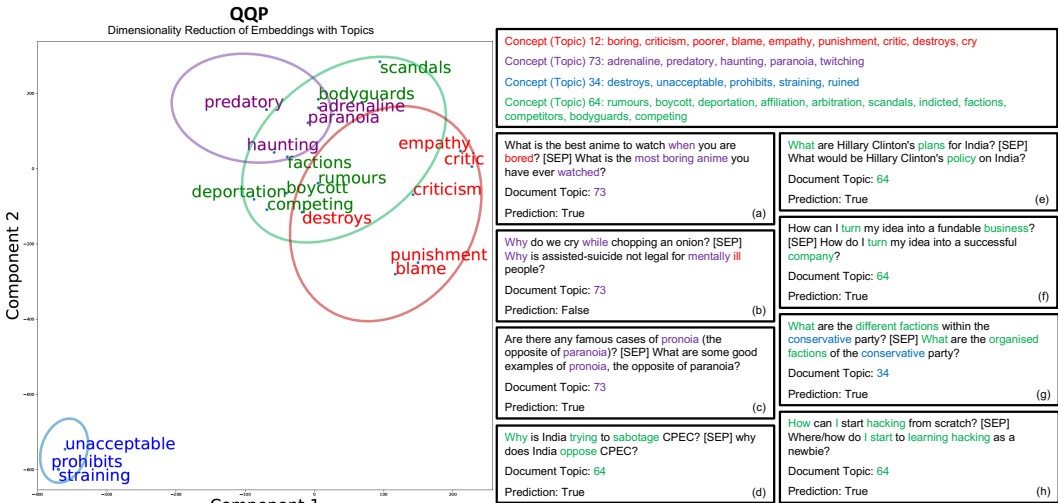

Figure 6: Visualization of CLDA's learned topics of contextual word embeddings. We show QQP's dataset-level interpretation with eight example documents.

the two questions are paraphrase of each other. For example, in Document (b) of Fig. 6, we can see that CLDA correctly interprets the model's predicted label 'False' with Concept 73 (Psychology). Similarly, in Document (e) of Fig. 6, we can see that CLDA correctly interprets the model's predicted label 'True' with Concept 64 (strategies).

**Word-Level Interpretations.** For word-level conceptual interpretations, we can observe that CLDA interprets PLM's prediction on Document (c) of Fig. 5 using words such as 'cat', 'floor', and 'garlic' that are related to household and daily life. Also, CLDA interprets PLM's prediction on Document (e) of Fig. 5 using words such as 'soldier' and 'border' that are related to national security. Similarly, for QQP's Document (d) (Fig. 6), CLDA correctly interprets the model prediction 'True' by identifying keywords such as 'sabotage' and 'oppose' with similar meanings in the topic of strategies. For QQP's Document (g), (Fig. 6), CLDA interprets the words in the both sentences with the same semantics, such as 'conservative' that is related to prevention and conservatives (note that in politics, 'conservative' refers to parties that tend to prevent/block new policies or legislation), and thereby predicting the correct label 'True'.

**Example Concepts.** Following Blei et al. (2003), we show the learned concepts on the RTE dataset in Table 3, which is complementary to aforementioned explanations. We select several different topics from Fig. 4 of the main paper. As in Sec. 4.2 of the main paper, we obtain top words from each concept via first calculating the average of the each word's corresponding contextual embeddings over the dataset, and then getting the nearest words to each topic center ($\boldsymbol{\mu}_i$) in the embedding space. As we can see in Table 3, CLDA can capture various concepts with profound and accurate semantics. Therefore, although PLM embeddings are contextual and continuous, our CLDA can still find conceptual patterns of words on the dataset-level.

# F THEORETICAL ANALYSIS ON CONTINUOUS WORD COUNTS

Before going to the claims and proofs, first we specify some basic problem settings and assumptions. Suppose there are $K + 1$ topic groups, each of which is regarded to be sampled from a parameterized multivariate Gaussian distribution. In specific, the $K + 1$'th distribution of topic has a much larger covariance, and in the same time, closed to the center of embedding space. The prementioned properties can be measured by a series of inequalities:

$$|\boldsymbol{\Sigma}_{K+1}| = \max_{i=1}^{k+1} |\boldsymbol{\Sigma}_i|. \tag{23}$$

The approximate marginal log-likelihood of word embeddings, i.e., the third term of the ELBO as mentioned in Eqn. 4 of the main paper, is:

$$\mathcal{L}^{(train)} = \sum_{j=1}^{L_m} \mathbb{E}_q[\log p(\mathbf{e}_{mj}|z_{mj}, \boldsymbol{\mu}_{z_{mj}}, \boldsymbol{\Sigma}_{z_{mj}})]$$
$$= \sum_{m,j,i} \phi_{mji} w_{mj} \{ -\tfrac{1}{2}(\mathbf{e}_{mj} - \boldsymbol{\mu}_i)^T \boldsymbol{\Sigma}_i^{-1}(\mathbf{e}_{mj} - \boldsymbol{\mu}_i) - \log[(2\pi)^{d/2}|\boldsymbol{\Sigma}_i|^{1/2}] \}. \quad (24)$$

The above equation is the training objective, yet for fair comparison of different training schemes, we calculate the approximated likelihood with word count 1 for all words.

$$\mathcal{L}^{(eval)} = \sum_{j=1}^{L_m} \mathbb{E}_q[\log p'(\mathbf{e}_{mj}|z_{mj}, \boldsymbol{\mu}_{z_{mj}}, \boldsymbol{\Sigma}_{z_{mj}})]$$
$$= \sum_{m,j,i} \phi_{mji} \{ -\tfrac{1}{2}(\mathbf{e}_{mj} - \boldsymbol{\mu}_i)^T \boldsymbol{\Sigma}_i^{-1}(\mathbf{e}_{mj} - \boldsymbol{\mu}_i) - \log[(2\pi)^{d/2}|\boldsymbol{\Sigma}_i|^{1/2}] \}. \quad (25)$$

### F.1 GAUSSIAN MIXTURE MODELS

Suppose we have a ground truth GMM model with parameters $\boldsymbol{\pi}^* \in \mathbb{R}^K$ and $\{\boldsymbol{\mu}_k^*, \boldsymbol{\Sigma}_k^*\}_{k=1}^K$, with $K$ different Gaussian distributions. In the dataset, let $N$ and $N_s$ denote the numbers of non-stop-words and stop-words, respectively. Then the marginal log likelihood of a learned GMM model on a given data sample $\mathbf{e}$ can be written as

$$p(\mathbf{e}|\{\boldsymbol{\mu}, \boldsymbol{\Sigma}\}, \boldsymbol{\pi}) = \sum_{k=1}^K \boldsymbol{\pi}_k \mathcal{N}(\mathbf{e}; \boldsymbol{\mu}_k, \boldsymbol{\Sigma}_k). \quad (26)$$

Assuming a dataset of $N + N_s$ words $\{\mathbf{e}_i\}_{i=1}^{N+N_s}$ and taking the associated weights $w_i$ for each word into account, the log-likelihood of the dataset can be written as

$$\sum_{i=1}^{N+N_s} p(\mathbf{e}_i|\{\boldsymbol{\mu}_k, \boldsymbol{\Sigma}_k\}_{k=1}^K, \boldsymbol{\pi}) = \sum_{i=1}^N \log \sum_{k=1}^K w_i \boldsymbol{\pi}_k \mathcal{N}(\mathbf{e}_i; \boldsymbol{\mu}_k, \boldsymbol{\Sigma}_k) + \sum_{i=N+1}^{N+N_s} \log \sum_{k=1}^K w_i \boldsymbol{\pi}_k \mathcal{N}(\mathbf{e}_i; \boldsymbol{\mu}_k, \boldsymbol{\Sigma}_k). \quad (27)$$

Leveraging Jensen's inequality, we obtain a lower bound of the above quantity (denoting as $\boldsymbol{\Theta}$ the collection of parameters $\{\boldsymbol{\mu}_k, \boldsymbol{\Sigma}_k\}_{k=1}^K$ and $\boldsymbol{\pi}$):

$$\mathcal{L}_{GMM}(\boldsymbol{\Theta}, \{w_i\}) = \sum_{i=1}^N w_i \log \sum_{k=1}^K \boldsymbol{\pi}_k \mathcal{N}(\mathbf{e}_i; \boldsymbol{\mu}_k, \boldsymbol{\Sigma}_k) + \sum_{i=N+1}^{N+N_s} w_i \log \sum_{k=1}^K \boldsymbol{\pi}_k \mathcal{N}(\mathbf{e}_i; \boldsymbol{\mu}_k, \boldsymbol{\Sigma}_k) + C, \quad (28)$$

where C is a constant.

In the following theoretical analysis, we consider the following three different configurations of the weights $w_i$.

**Definition F.1 (Weight Configurations).** We define three different weight configurations as follows:

- Identical Weights: $w_i = \frac{1}{N+N_s}, i \in \{1, 2, \ldots, N + N_s\}$

- Ground-Truth Weights : $w_i = \begin{cases} \frac{1}{N}, & i \in \{1, 2, \ldots, N\} \\ 0, & i \in \{N+1, N+2, \ldots, N+N_s\} \end{cases}$

- Attention-Based Weights: $w_i = \begin{cases} \lambda_1 \in [\frac{1}{N+N_s}, \frac{1}{N}], & i \in \{1, 2, \ldots, N\} \\ \lambda_2 \in [0, \frac{1}{N+N_s}], & i \in \{N+1, N+2, \ldots, N+N_s\} \end{cases}$

**Definition F.2 (Advanced Weight Configurations).** We define three different weight configurations as follows:

- Identical Weights: $w_i = \frac{1}{N+N_s}, i \in \{1, 2, \ldots, N + N_s\}$

- Ground-Truth Weights : $w_i = \begin{cases} \frac{1}{N}, & i \in \{1, 2, \ldots, N\} \\ 0, & i \in \{N+1, N+2, \ldots, N+N_s\} \end{cases}$

- Attention-Based Weights: $w_i \in \begin{cases} [\frac{1}{N+N_s}, \frac{1}{N}], & i \in \{1, 2, \ldots, N\} \\ [0, \frac{1}{N+N_s}], & i \in \{N+1, N+2, \ldots, N+N_s\} \end{cases}$

**Definition F.3** (**Optimal Parameters**). With Definition F.1, the corresponding optimal parameters are then defined as follows:

$$\mathbf{\Theta}_I = \arg\max_{\mathbf{\Theta}} \mathcal{L}(\mathbf{\Theta}; \mathbf{w} \to \text{Identical}), \tag{29}$$

$$\mathbf{\Theta}_G = \arg\max_{\mathbf{\Theta}} \mathcal{L}(\mathbf{\Theta}; \mathbf{w} \to \text{GT}), \tag{30}$$

$$\mathbf{\Theta}_A = \arg\max_{\mathbf{\Theta}} \mathcal{L}(\mathbf{\Theta}; \mathbf{w} \to \text{Attention}), \tag{31}$$

where $\mathbf{w} \to$ Identical, $\mathbf{w} \to$ GT, and $\mathbf{w} \to$ Attention indicates that 'Identical Weights', 'Ground-Truth Weights', and 'Attention-Based Weights' are used, respectively.

**Lemma F.4.** *Suppose we have two series of functions $\{f_{1,i}(x)\}$ and $\{f_{2,i}(x)\}$, with two non-negative weighting parameters $\lambda_1, \lambda_2$ satisfying $N\lambda_1 + N_s\lambda_2 = 1$. We define the final objective function $f(\cdot)$ as:*

$$f(x; \lambda_1, \lambda_2) = \lambda_1 \sum_{i=1}^{N} f_{1,i}(x) + \lambda_2 \sum_{i=N+1}^{N_s} f_{2,i}(x). \tag{32}$$

*We assume two pairs of parameters $(\lambda_1, \lambda_2)$ and $(\lambda_1', \lambda_2')$, where*

$$\lambda_1 \geq \lambda_1', \tag{33}$$

$$\lambda_2 \leq \lambda_2'. \tag{34}$$

*Defining the optimal values of the objective function for different weighting parameters as*

$$\widehat{x} = \arg\max_{x} f(x; \lambda_1, \lambda_2), \tag{35}$$

$$\widehat{x}' = \arg\max_{x} f(x; \lambda_1', \lambda_2'), \tag{36}$$

*we then have that*

$$f(\widehat{x}; \tfrac{1}{N}, 0) \geq f(\widehat{x}'; \tfrac{1}{N}, 0). \tag{37}$$

*Proof.* We prove this theorem by contradiction. Suppose that we have

$$f(\widehat{x}; \tfrac{1}{N}, 0) < f(\widehat{x}'; \tfrac{1}{N}, 0). \tag{38}$$

According to Eqn. 49, i.e., $\lambda_1 \geq \lambda_1'$, and the equation $N\lambda_1 + N_s\lambda_2 = 1$, we have

$$\lambda_1\lambda_2' = \lambda_1 \frac{1-N\lambda_1'}{N_s} \geq \lambda_1' \frac{1-N\lambda_1}{N_s} = \lambda_1'\lambda_2. \tag{39}$$

According to Eqn. 36, we have the following equality:

$$f(\widehat{x}; \lambda_1', \lambda_2') \leq f(\widehat{x}'; \lambda_1', \lambda_2'). \tag{40}$$

Combined with the aforementioned assumption in Eqn. 38, we have that

$$\lambda_2' f(\widehat{x}; \lambda_1, \lambda_2) = \lambda_1\lambda_2' \sum_{i=1}^{N} f_{1,i}(\widehat{x}) + \lambda_2\lambda_2' \sum_{i=N+1}^{N_s} f_{2,i}(\widehat{x}) \tag{41}$$

$$= (\lambda_1'\lambda_2 \sum_{i=1}^{N} f_{1,i}(\widehat{x}) + \lambda_2'\lambda_2 \sum_{i=N+1}^{N_s} f_{2,i}(\widehat{x})) + (N(\lambda_1\lambda_2' - \lambda_1'\lambda_2) \cdot \tfrac{1}{N} \sum_{i=1}^{N} f_{1,i}(\widehat{x})) \tag{42}$$

$$= \lambda_2 f(\widehat{x}; \lambda_1', \lambda_2') + N(\lambda_1\lambda_2' - \lambda_1'\lambda_2) f(\widehat{x}; \tfrac{1}{N}, 0) \tag{43}$$

$$< \lambda_2 f(\widehat{x}'; \lambda_1', \lambda_2') + N(\lambda_1\lambda_2' - \lambda_1'\lambda_2) f(\widehat{x}'; \tfrac{1}{N}, 0) \tag{44}$$

$$= (\lambda_1'\lambda_2 \sum_{i=1}^{N} f_{1,i}(\widehat{x}') + \lambda_2'\lambda_2 \sum_{i=N+1}^{N_s} f_{2,i}(\widehat{x}')) + (N(\lambda_1\lambda_2' - \lambda_1'\lambda_2) \cdot \tfrac{1}{N} \sum_{i=1}^{N} f_{1,i}(\widehat{x}')) \tag{45}$$

$$= \lambda_1\lambda_2' \sum_{i=1}^{N} f_{1,i}(\widehat{x}') + \lambda_2\lambda_2' \sum_{i=N+1}^{N_s} f_{2,i}(\widehat{x}') \tag{46}$$

$$= \lambda_2' f(\widehat{x}'; \lambda_1, \lambda_2), \tag{47}$$

which contradicts the definition of $\widehat{x}$ in Eqn. 35 (i.e., $\widehat{x}$ maximizes $f(x; \lambda_1, \lambda_2)$), completing the proof. $\qquad\square$

**Lemma F.5.** *Suppose we have two series of functions $\{f_{1,i}(x)\}$ and $\{f_{2,i}(x)\}$, with two series of non-negative weighting parameters $\boldsymbol{\lambda}_1 = [\lambda_{1,i}]_{i=1}^N, \boldsymbol{\lambda}_2 = [\lambda_{2,i}]_{i=N+1}^{N_s}$ satisfying $\sum_{i=1}^N \lambda_{1,i} + \sum_{i=N+1}^{N_s} \lambda_{2,i} = 1$. We define the final objective function $f(\cdot)$ as:*

$$f(x; \boldsymbol{\lambda}_1, \boldsymbol{\lambda}_2) = \sum_{i=1}^N \lambda_{1,i} f_{1,i}(x) + \sum_{i=N+1}^{N_s} \lambda_{2,i} f_{2,i}(x). \tag{48}$$

*We assume two pairs of parameters $(\boldsymbol{\lambda}_1, \boldsymbol{\lambda}_2)$ and $(\boldsymbol{\lambda}_1', \boldsymbol{\lambda}_2')$, where*

$$\lambda_{1,i} \geq \lambda_{1,i}', \quad i \in \{1, 2, ..., N\}, \tag{49}$$
$$\lambda_{2,i} \leq \lambda_{2,i}', \quad i \in \{N+1, N+2, ..., N_s\}. \tag{50}$$

*Defining the optimal values of the objective function for different weighting parameters as*

$$\widehat{x} = \arg\max_x f(x; \boldsymbol{\lambda}_1, \boldsymbol{\lambda}_2), \tag{51}$$
$$\widehat{x}' = \arg\max_x f(x; \boldsymbol{\lambda}_1', \boldsymbol{\lambda}_2'), \tag{52}$$
$$x^* = \arg\max f(x, \tfrac{1}{N}, \mathbf{0}). \tag{53}$$

*Under the following **Assumptions** (with $\mathbf{1}$ and $\mathbf{0}$ denoting vectors with all entries equal to $1$ and $0$, respectively):*

1. *$f(\widehat{x}, \mathbf{0}, \boldsymbol{\lambda}_2) \leq f(\widehat{x}', \mathbf{0}, \boldsymbol{\lambda}_2)$.*

2. *$f(x; \boldsymbol{\lambda}, \mathbf{0}) \geq f(x'; \boldsymbol{\lambda}, \mathbf{0})$, iff $\|x - x^*\| \leq \|x' - x^*\|$, $\boldsymbol{\lambda} \geq 0$, $\|\boldsymbol{\lambda}\|_1 = 1$.*

*we have that*

$$f(\widehat{x}; \tfrac{1}{N}, \mathbf{0}) \geq f(\widehat{x}'; \tfrac{1}{N}, \mathbf{0}). \tag{54}$$

*Proof.* We start with proving the following equality by contradiction:

$$\|\widehat{x} - x^*\| \leq \|\widehat{x}' - x^*\|. \tag{55}$$

Specifically, if

$$\|\widehat{x} - x^*\| > \|\widehat{x}' - x^*\|, \tag{56}$$

leveraging the Assumption 1 and 2 above, we have that

$$f(\widehat{x}; \boldsymbol{\lambda}_1, \boldsymbol{\lambda}_2) = f(\widehat{x}; \boldsymbol{\lambda}_1, \mathbf{0}) + f(\widehat{x}; \mathbf{0}, \boldsymbol{\lambda}_2) < f(\widehat{x}'; \boldsymbol{\lambda}_1, \mathbf{0}) + f(\widehat{x}'; \mathbf{0}, \boldsymbol{\lambda}_2) = f(\widehat{x}'; \boldsymbol{\lambda}_1, \boldsymbol{\lambda}_2), \tag{57}$$

which contradicts Eqn. 51. Therefore, Eqn. 55 holds.

Combining Eqn. 55 and Assumption 2 above, we have that

$$f(\widehat{x}; \tfrac{1}{N}, \mathbf{0}) \geq f(\widehat{x}'; \tfrac{1}{N}, \mathbf{0}), \tag{58}$$

concluding the proof. $\qquad\square$

Based on the definitions and lemmas above, we have the following theorems:

**Theorem F.6** (**Advantage of $\Theta_A$ in the Simplified Case**). *With Definition F.1 and Definition F.3, comparing $\Theta_I$, $\Theta_G$, and $\Theta_A$ by evaluating them on the marginal log-likelihood of non-stop-words, i.e., $\mathcal{L}(\cdot, w \to GT)$, we have that*

$$\mathcal{L}_{GMM}(\boldsymbol{\Theta}_I; \mathbf{w} \to GT) \leq \mathcal{L}_{GMM}(\boldsymbol{\Theta}_A; \mathbf{w} \to GT) \leq \mathcal{L}_{GMM}(\boldsymbol{\Theta}_G; \mathbf{w} \to GT). \tag{59}$$

*Proof.* First, by definition one can easily find that $\boldsymbol{\Theta}_G$ achieves the largest $\mathcal{L}(\cdot; \mathbf{w} \to \text{GT})$ among the three:

$$\max[\mathcal{L}_{GMM}(\boldsymbol{\Theta}_I; \mathbf{w} \to \text{GT}), \mathcal{L}_{GMM}(\boldsymbol{\Theta}_A; \mathbf{w} \to \text{GT})] \leq \max_{\boldsymbol{\Theta}} \mathcal{L}_{GMM}(\boldsymbol{\Theta}; \mathbf{w} \to \text{GT}) = \mathcal{L}_{GMM}(\boldsymbol{\Theta}_G; \mathbf{w} \to \text{GT}). \quad (60)$$

Next, we set $\{w_i\}_{i=1}^N$ to $\lambda_1$ and $\{w_i\}_{i=N+1}^{N+N_s}$ to $\lambda_2$, respectively; we rewrite $\log \sum_{k=1}^K \pi_k \mathcal{N}(\mathbf{e}_i; \boldsymbol{\mu}_k, \boldsymbol{\Sigma}_k)$ as $f_{1,i}(x)$ for $i \in \{1, 2, \ldots, N\}$ and $f_{2,i}(x)$ for $i \in \{N+1, N+1, \ldots, N+N_s\}$, where $x$ corresponds to $\boldsymbol{\Theta} \triangleq (\boldsymbol{\pi}, \{\boldsymbol{\mu}_k, \boldsymbol{\Sigma}_k\}_{k=1}^K)$. By Lemma F.4, we have that

$$\mathcal{L}_{GMM}(\boldsymbol{\Theta}_A; \mathbf{w} \to \text{GT}) \leq \mathcal{L}_{GMM}(\boldsymbol{\Theta}_G; \mathbf{w} \to \text{GT}). \quad (61)$$

Combining Eqn. 60 and Eqn. 61 concludes the proof. $\qquad\square$

Theorem F.7 shows that under mild assumptions, the attention-based weights can help produce better estimates of $\boldsymbol{\Theta}$ in the presence of noisy stop-words and therefore learns higher-quality topics from the corpus, improving both generalization performance and interpretability of PLMs.

**Theorem F.7 (Advantage of $\boldsymbol{\Theta}_A$ in the General Case).** *With Definition F.2 and Definition F.3, comparing $\boldsymbol{\Theta}_I$, $\boldsymbol{\Theta}_G$, and $\boldsymbol{\Theta}_A$ by evaluating them on the marginal log-likelihood of non-stop-words, i.e., $\mathcal{L}_{GMM}(\cdot, w \to GT)$, we have that*

$$\mathcal{L}_{GMM}(\boldsymbol{\Theta}_I; \mathbf{w} \to GT) \leq \mathcal{L}_{GMM}(\boldsymbol{\Theta}_A; \mathbf{w} \to GT) \leq \mathcal{L}_{GMM}(\boldsymbol{\Theta}_G; \mathbf{w} \to GT). \quad (62)$$

*Proof.* First, by definition one can easily find that $\boldsymbol{\Theta}_G$ achieves the largest $\mathcal{L}(\cdot; \mathbf{w} \to \text{GT})$ among the three:

$$\max[\mathcal{L}_{GMM}(\boldsymbol{\Theta}_I; \mathbf{w} \to \text{GT}), \mathcal{L}_{GMM}(\boldsymbol{\Theta}_A; \mathbf{w} \to \text{GT})] \leq \max_{\boldsymbol{\Theta}} \mathcal{L}_{GMM}(\boldsymbol{\Theta}; \mathbf{w} \to \text{GT}) = \mathcal{L}_{GMM}(\boldsymbol{\Theta}_G; \mathbf{w} \to \text{GT}). \quad (63)$$

Next, we invoke Lemma F.5 by (1) setting $\{w_i\}_{i=1}^N$ to $\boldsymbol{\lambda}_1$ and $\{w_i\}_{i=N+1}^{N+N_s}$ to $\boldsymbol{\lambda}_2$, respectively, and (2) rewriting $\log \sum_{k=1}^K \pi_k \mathcal{N}(\mathbf{e}_i; \boldsymbol{\mu}_k, \boldsymbol{\Sigma}_k)$ as $f_{1,i}(x)$ for $i \in \{1, 2, \ldots, N\}$ and $f_{2,i}(x)$ for $i \in \{N+1, N+1, \ldots, N+N_s\}$, where $x$ corresponds to $\boldsymbol{\Theta} \triangleq (\boldsymbol{\pi}, \{\boldsymbol{\mu}_k, \boldsymbol{\Sigma}_k\}_{k=1}^K)$. By Lemma F.5, we then have that

$$\mathcal{L}_{GMM}(\boldsymbol{\Theta}_A; \mathbf{w} \to \text{GT}) \leq \mathcal{L}_{GMM}(\boldsymbol{\Theta}_G; \mathbf{w} \to \text{GT}). \quad (64)$$

Note that because $f_{1,i}(\cdot)$ and $f_{2,i}(\cdot)$ are Gaussian, therefore Assumption 1 and 2 in Lemma F.5 hold naturally under mild regularity conditions.

Combining Eqn. 70 and Eqn. 71 concludes the proof. $\qquad\square$

## F.2 CLDA AS INTERPRETERS

As mentioned in Eqn. 4 of the main paper, the ELBO of the marginal likelihood (denoting as $\boldsymbol{\Theta}$ the collection of parameters $\phi, \boldsymbol{\gamma}$ and $\{\boldsymbol{\mu}_k, \boldsymbol{\Sigma}_k\}_{k=1}^K$) is as follows:

$$\begin{aligned} \mathcal{L}_{CLDA}(\boldsymbol{\Theta}; \{w_i\}) &= \sum_{j=1}^{L'} \mathbb{E}_q[\log p(\mathbf{e}_{mj}|z_{mj}, \boldsymbol{\mu}_{z_{mj}}, \boldsymbol{\Sigma}_{z_{mj}})] \\ &= \sum_{m,j} w_{mj} \sum_i \phi_{mji}\{-\tfrac{1}{2}(\mathbf{e}_{mj} - \boldsymbol{\mu}_i)^T \boldsymbol{\Sigma}_i^{-1}(\mathbf{e}_{mj} - \boldsymbol{\mu}_i) - \log[(2\pi)^{d/2}|\boldsymbol{\Sigma}_i|^{1/2}]\}. \end{aligned} \quad (65)$$

Based on the definitions and lemmas above, we have the following theorems:

**Theorem F.8 (Advantage of $\boldsymbol{\Theta}_A$ in the Simplified Case).** *With Definition F.1 and Definition F.3, comparing $\boldsymbol{\Theta}_I$, $\boldsymbol{\Theta}_G$, and $\boldsymbol{\Theta}_A$ by evaluating them on the marginal log-likelihood of non-stop-words, i.e., $\mathcal{L}(\cdot, w \to GT)$, we have that*

$$\mathcal{L}_{CLDA}(\boldsymbol{\Theta}_I; \mathbf{w} \to GT) \leq \mathcal{L}_{CLDA}(\boldsymbol{\Theta}_A; \mathbf{w} \to GT) \leq \mathcal{L}_{CLDA}(\boldsymbol{\Theta}_G; \mathbf{w} \to GT). \quad (66)$$

*Proof.* First, by definition one can easily find that $\boldsymbol{\Theta}_G$ achieves the largest $\mathcal{L}(\cdot; \mathbf{w} \to \text{GT})$ among the three:

$$\max[\mathcal{L}_{CLDA}(\boldsymbol{\Theta}_I; \mathbf{w} \to \text{GT}), \mathcal{L}_{CLDA}(\boldsymbol{\Theta}_A; \mathbf{w} \to \text{GT})] \leq \max_{\boldsymbol{\Theta}} \mathcal{L}_{CLDA}(\boldsymbol{\Theta}; \mathbf{w} \to \text{GT}) = \mathcal{L}_{CLDA}(\boldsymbol{\Theta}_G; \mathbf{w} \to \text{GT}). \quad (67)$$

Next, we set $\cup_m \{w_{mj}\}_{j=1}^{N_m}$ to $\boldsymbol{\lambda}_1$ and $\cup_m \{w_{mj}\}_{j=N_m+1}^{N_m+N_{m,s}}$ to $\boldsymbol{\lambda}_2$, respectively; we rewrite $\sum_i \phi_{mji}\{-\frac{1}{2}(\mathbf{e}_{mj} - \boldsymbol{\mu}_i)^T \boldsymbol{\Sigma}_i^{-1}(\mathbf{e}_{mj} - \boldsymbol{\mu}_i) - \log[(2\pi)^{d/2}|\boldsymbol{\Sigma}_i|^{1/2}]\}$ as $f_{1,j}(x)$ for $j \in \cup_m\{1, 2, \ldots, N_m\}$ and $f_{2,j}(x)$ for $j \in \cup_m\{N_m + 1, N_m + 1, \ldots, N_m + N_{m,s}\}$, where $x$ corresponds to $\boldsymbol{\Theta} \triangleq (\boldsymbol{\phi}, \boldsymbol{\gamma}, \{\boldsymbol{\mu}_k, \boldsymbol{\Sigma}_k\}_{k=1}^K)$. By Lemma F.4, we have that

$$\mathcal{L}_{CLDA}(\boldsymbol{\Theta}_A; \mathbf{w} \to \text{GT}) \leq \mathcal{L}_{CLDA}(\boldsymbol{\Theta}_G; \mathbf{w} \to \text{GT}). \tag{68}$$

Combining Eqn. 67 and Eqn. 68 concludes the proof. $\qquad\square$

Theorem F.8 shows that under mild assumptions, the attention-based weights can help produce better estimates of $\boldsymbol{\Theta}$ in the presence of noisy stop-words and therefore learns higher-quality topics from the corpus, improving both generalization performance and interpretability of PLMs.

**Theorem F.9** (**Advantage of $\boldsymbol{\Theta}_A$ in the General Case**). *With Definition F.2 and Definition F.3, comparing $\boldsymbol{\Theta}_I$, $\boldsymbol{\Theta}_G$, and $\boldsymbol{\Theta}_A$ by evaluating them on the marginal log-likelihood of non-stop-words, i.e., $\mathcal{L}_{CLDA}(\cdot, w \to GT)$, we have that*

$$\mathcal{L}_{CLDA}(\boldsymbol{\Theta}_I; \mathbf{w} \to GT) \leq \mathcal{L}_{CLDA}(\boldsymbol{\Theta}_A; \mathbf{w} \to GT) \leq \mathcal{L}_{CLDA}(\boldsymbol{\Theta}_G; \mathbf{w} \to GT). \tag{69}$$

*Proof.* First, by definition one can easily find that $\boldsymbol{\Theta}_G$ achieves the largest $\mathcal{L}(\cdot; \mathbf{w} \to \text{GT})$ among the three:

$$\max[\mathcal{L}_{CLDA}(\boldsymbol{\Theta}_I; \mathbf{w} \to \text{GT}), \mathcal{L}_{CLDA}(\boldsymbol{\Theta}_A; \mathbf{w} \to \text{GT})] \leq \max_{\boldsymbol{\Theta}} \mathcal{L}_{CLDA}(\boldsymbol{\Theta}; \mathbf{w} \to \text{GT}) = \mathcal{L}_{CLDA}(\boldsymbol{\Theta}_G; \mathbf{w} \to \text{GT}). \tag{70}$$

Next, we invoke Lemma F.5 by (1) setting $\cup_m \{w_{mj}\}_{j=1}^{N_m}$ to $\boldsymbol{\lambda}_1$ and $\cup_m \{w_{mj}\}_{j=N_m+1}^{N_m+N_{m,s}}$ to $\boldsymbol{\lambda}_2$, respectively, and (2) rewriting $\sum_i \phi_{mji}\{-\frac{1}{2}(\mathbf{e}_{mj} - \boldsymbol{\mu}_i)^T \boldsymbol{\Sigma}_i^{-1}(\mathbf{e}_{mj} - \boldsymbol{\mu}_i) - \log[(2\pi)^{d/2}|\boldsymbol{\Sigma}_i|^{1/2}]\}$ as $f_{1,j}(x)$ for $j \in \cup_m\{1, 2, \ldots, N_m\}$ and $f_{2,j}(x)$ for $j \in \cup_m\{N_m + 1, N_m + 1, \ldots, N_m + N_{m,s}\}$, where $x$ corresponds to $\boldsymbol{\Theta} \triangleq (\boldsymbol{\phi}, \boldsymbol{\gamma}, \{\boldsymbol{\mu}_k, \boldsymbol{\Sigma}_k\}_{k=1}^K)$. By Lemma F.5, we then have that

$$\mathcal{L}_{CLDA}(\boldsymbol{\Theta}_A; \mathbf{w} \to \text{GT}) \leq \mathcal{L}_{CLDA}(\boldsymbol{\Theta}_G; \mathbf{w} \to \text{GT}). \tag{71}$$

Note that because $f_{1,j}(\cdot)$ and $f_{2,j}(\cdot)$ are very close to Gaussian, therefore Assumption 1 and 2 in Lemma F.5 hold naturally under mild regularity conditions.

Combining Eqn. 70 and Eqn. 71 concludes the proof. $\qquad\square$

