# OpenReview forum: "Intepreting & Improving Pretrained Language Models: A Probabilistic Conceptual Approach"
_ICLR.cc/2023/Conference — Submitted to ICLR 2023_

### Official Review · Reviewer_UZmn · 2022-10-24

**Confidence:** 3
**Correctness:** 4
**Technical Novelty And Significance:** 4
**Empirical Novelty And Significance:** 3
**Recommendation:** 8

**Clarity, Quality, Novelty And Reproducibility:**

The work is clear and of good quality. The method is novel and there are sufficient details for reproducibility.

**Strength And Weaknesses:**

Strength:
- Novel method of training CLDA jointly with LMs
- Interpreting method as a bonus, strong theoretical backing
- Good results on GLUE benchmark

Weaknesses:
- No major weaknesses

Notes:
- Concept 67 references "West Asia" and not necessarily "Islam" as per the highlighted red words. For example, "Pasha" and "Qaeda" are not related to Islam.

**Summary Of The Paper:**

The authors propose to apply Continuous LDA to pre-trained language models. The authors outline how to jointly train in a weighted loss function both the CLDA and the LM, making CLDA act as a regularizer for language models. In addition, CLDA can be used to interpret LMs, by using attention weights to group words at a concept and document level. The authors finally show that the CLDA actually improves the performance of LMs on the GLUE benchmark, particularly BERT-based ones.

**Summary Of The Review:**

The method introduced in the paper proposes to train an LM jointly with CLDA, shows how to interpret the resulting CLDA that uses the LM's attention weights, and improves performance on the GLUE benchmark. These contributions are solid and benefit the community.

---

> ### Author Response · Authors · 2022-11-09
> **To Reviewer UZmn**
>
> Thank you for your encouraging comments! We are glad that you find our work ``"clear and of good quality"``, and our contributions ``"solid and benefit the community"``. Also, we are grateful for your notes on the interpretations of concepts, and will address the underlining issues in the revision.

---

### Official Review · Reviewer_YUbf · 2022-10-24

**Confidence:** 4
**Correctness:** 3
**Technical Novelty And Significance:** 2
**Empirical Novelty And Significance:** 2
**Recommendation:** 3

**Clarity, Quality, Novelty And Reproducibility:**

The paper is generally well written and easy to follow. The CLDA algorithm is mathematically sound. The method should be reproducible if the author can public their training and inference code.

**Strength And Weaknesses:**

Strength: The proposed CLDA model is interesting and new as far as I know. The inference and learning algorithms are well designed.

Weakness: The motivation of this paper is somehow weak to me. Why is it necessary to build LDA models upon the PLMs? What can we gain from the interpretation results such as those shown in Figure 4? How can we evaluate them? For example, if a PLM made wrong prediction on certain example, will the CLDA model give the interpretation and help to fix the error? The application scenarios of the CLDA (from the interpretation view) are completely missed in the paper.
For the quantitative results part, it is less discussed what makes the improvement over the baseline models. If it is the regularization effect, is it possible that the baseline models can be fine-tuned with finer but simpler regularization to achieve the similar results, considering that the bi-level optimization of PLM and CLDA is much more inefficient?
Besides, why not include all the GLUE tasks rather than selecting part of them? And all the experiments and analysis are only tested in fine-tuning stage, it is unclear whether we can apply it in pretraining and what the results could be.


**Summary Of The Paper:**

This paper proposes adding topic model upon pretrained language models (PLMs), in order to increase its interpretability and downstream performance. More specifically, the authors design CLDA, a new LDA model compatible with continuous word counts, which are computed via aggregated attention scores of the last layer in PLM. Each topics/concept is modeled as Gaussian distribution in the contextual embedding space. The CLDA can be used for interpretation of the PLM outputs when decoupled with PLM training, or used for regularization when jointly trained with PLMs. Experiments shows that CLDA can distinguish topics from word level to document level, and improve several GLUE tasks with joint learning.

**Summary Of The Review:**

The paper proposes a LDA topic model based on the PLM, which can serve as interpreter and regularizor of the PLM. The method is technically sound. However, the paper is lack of discussion about how the interpretation results can be used and evaluated. The experimental results are incomplete and not that convincing, which prevent further exploration of the proposed method.

---

> ### Author Response · Authors · 2022-11-09
> **To Reviewer YUbf**
>
> **Q1: "The motivation of this paper is somehow weak to me."**
>
> Primarily, in both the main paper and Appendix, we explained the correlation between the conceptual explanation and model prediction via multiple examples, on the three levels of interpretations. Furthermore, we provide cases of right and wrong prediction of the PLMs, and provide the interpretation from the CLDA module.
>
> **Q2: "For the quantitative results part, it is less discussed what makes the improvement over the baseline models."**
>
> As we discussed in the prementioned statements, the justification of our model as a good regulator can be verify by the specific experimental settings. Also, we **do** conducted ablation study in the paper, and the results reveal that the attention-based continuous word count is a critical design to boost our model performance. For the time limit, we do not conduct experiments on all of the GLUE task, and will also dive into the pretraining stage in the future work.

---

### Official Review · Reviewer_iDsq · 2022-10-24

**Confidence:** 4
**Correctness:** 3
**Technical Novelty And Significance:** 2
**Empirical Novelty And Significance:** 1
**Recommendation:** 3

**Clarity, Quality, Novelty And Reproducibility:**

Clarity: Poor
Quality: OK
Novelty: Poor
Reproducibility: Ok

**Strength And Weaknesses:**

Strengthes:
1. The framework is technically sound and the adaptation of conventional topic model to pre-trained language models is interesting and can be helpful.
2. The authors conduct both theoretical and empirical analysis of the proposed method.

Weaknesses:
1. The paper is not very well written and some parts of it are quite hard to follow/parse. Also, the citation style is strange and not reader-friendly.
2. While the framework of  topic model with pre-trained language model is technically correct, the main claims of the paper (i.e., it can serve as good interpreter and regulator) are not well supported. First, it is unclear how finding the concepts and topics of different level will help users interpret the model's prediction. From this point of view, the proposed method is not as good as (or better) than existing interpretation methods for pre-trained models using either attention weights or shapley values. Second, for the regularization effect, the results on the GLUE benchmark in Table 1 is very poor compared to that in literature. For example, BERT-base can easily achieve around 57 for CoLA, 88 for MRPC in F1, and 66 in RTE, as well as on all other datasets. This is very different from the baseline results reported in Table 1. The experimental settings are also not very clearly described, making the effectiveness of the proposed method as regulator not convincing.
3. There already exists several works on extending topic models with pre-trained models [1-2]. The authors do not discuss the contribution with respect to these works, making it hard to judge the actual technical contribution of the proposed framework.

[1] BERTopic: Neural topic modeling with a class-based TF-IDF procedure
[2] Topic Modeling with Contextualized Word Representation Clusters

**Summary Of The Paper:**

This paper extend conventional topic model methods to contextualized word embeddings from BERT-like pre-trained models. The authors propose several modifications including using attention weights as continuous word counts and modeling and considering a contextualized word embeddings to be drawn from a Gaussian distribution corresponding to its latent topic. The authors propose to use the modified topic model as either interpreter for improved interpretability and as a regulator for improved performance.

**Summary Of The Review:**

This paper extends topic models in PLMs and uses it as interpreter and regulator. However the experiments does not well support the main claims and the novelty of the technical contribution with respect to similar literature is unclear.

---

> ### Author Response · Authors · 2022-11-09
> **To Reviewer iDsq**
>
> **Q1: "The paper is not very well written and some parts of it are quite hard to follow/parse."**
>
> We are sorry for the confusion. We will refine the writing of the paper, and improve the readability of certain parts, like the conceptual interpretation results. For the citation styles, we will double check and strictly follow the ICLR instructions.
>
> **Q2: "the main claims of the paper (i.e., it can serve as good interpreter and regulator) are not well supported."**
>
> Primarily, in both the main paper and Appendix, we explained the correlation between the conceptual explanation and model prediction via multiple examples, on the three levels of interpretations. Also,as we discussed in the prementioned statements, the justification of our model as a good regulator can be verify by the specific experimental settings.
>
> **Q3:"There already exists several works on extending topic models with pre-trained models"**
>
> Thank you for pointing us to the interesting references. We will include reference in the revision. As we discussed in the prementioned statements, our model is methodologically **different** from the existing attention-based explanations and contextual topic models.

---

### Official Review · Reviewer_c7kD · 2022-10-25

**Confidence:** 3
**Correctness:** 2
**Technical Novelty And Significance:** 2
**Empirical Novelty And Significance:** 2
**Recommendation:** 3

**Clarity, Quality, Novelty And Reproducibility:**

It is difficult to understand the paper's goals and motivations, as well as some of its terminology.

**Strength And Weaknesses:**

Strengths:
* The paper follows the interesting idea of fitting a hierarchical Bayesian model to contextual word embedding.
* It features a detailed derivation of the probabilistic model and the variational inference algorithm.
* The probabilistic model can be used as a regulator during fine-tuning.

Weaknesses:
* The fine-tuned BERT-base and RoBERTa- base baselines in Table 1 have consistently worse performance than the official figures from their respective papers, casting doubt on the experimental setting in this paper. Additionally, it is strange that their RoBERTa-base model performs so much worse for QQP than BERT-base.
* The focus of the method is unclear: Are we looking for “interpretability” in the sense of (a) explaining a model’s decision boundary, (b) feature attribution for a particular model predictions, or (c) analyzing downstream corpora (e.g., like a topic model)? The authors seem to promise a mix of (a) and (b), but I can’t see how the inferred concepts help to give faithful or even plausible explanations. I would suggest that the authors feature a case study with user requirements & expectations about “interpretability” and then compare CLDA’s interpretations to the baselines. For example, claims of “improving readability and intuitiveness” need to be supported more explicitly by the experiments.
* The authors should be clear about the target setting. The introduction and method only use the term pre-trained language models (PLMs), while in the experiments the method is applied to LMs fine-tuned on sentence classification datasets.
* The paper should discuss the literature on attention-based explanations in BERT, see Bibal et al., 2022, “Is Attention Explanation? An Introduction to the Debate”, as well as other works that use contextualized embeddings for topic modeling, e.g., Grootendorst, 2022, “BERTopic: Neural topic modeling with a class-based TF-IDF procedure” and Bianchi et al., 2021, “Pre-training is a Hot Topic: Contextualized Document Embeddings Improve Topic Coherence”.


**Summary Of The Paper:**

The authors propose to interpret contextual word embeddings by fitting a Gaussian mixture model with a Dirichlet prior over documents (CLDA) to the embeddings. They apply this framework to analyze the embeddings of fine-tuned language models on a number of tasks, and show that the model can be used as a regulator during fine-tuning.

**Summary Of The Review:**

This paper proposes a novel interpretability approach using a Bayesian model fit to word embeddings. There are concerns about the experimental settings and intended use cases.

---

> ### Author Response · Authors · 2022-11-09
> **To Reviewer c7kD**
>
> **Q1: "The fine-tuned BERT-base and RoBERTa- base baselines in Table 1 have consistently worse performance than the official figures from their respective papers"**
>
> **(1)** To ensure fair comparison, during fine-tuning, we select the epoch for both baselines and CLDA entirely based on **validation** accuracy and report the **test** accuracy; in contrast, [5] directly chooses the **3rd epoch**; we argue that this is not rigorous and potentially "overfits" test sets.
>
> **(2)** We follow the convention of topic models and preprocess the documents into lower-case words for both baselines and CLDA; in contrast, [5] keeps the words unchanged.
>
> **(3)** Note that our CLDA can interpret **any** PLMs **without accuracy sacrifice**; therefore the exact accuracy for BERT-base is less relevant in our case.
>
> Therefore, our setting is different from original papers.
>
> We will also run both the baselines and our CLDA in the non-lower-case token setting, include the results in the revision, and update the rebuttal if we could make it before the discussion phase ends.
>
>
> **Q2: "I can’t see how the inferred concepts help to give faithful or even plausible explanations"**
>
> As illustrated in both the paper and the Appendix, our model can naturally provide predictions with Concept-Document-Word level interpretations in a hierarchical manner. Also, Thank you for the suggestion of "a case study with user requirements & expectations about ``'interpretability'`` and then compare CLDA’s interpretations to the baselines". We will conduct user study compared to the baseline methods, and discuss the results in the revision.
>
> **Q3:"The authors should be clear about the target setting."**
>
> As discussed in the Experimental Section of the paper, as well as the Appendix, our empirical results are based on the tasks in GLUE benchmark, which are all classification tasks. Basically, PLMs are not pretrained on the sentence classification tasks. However, these tasks (such as GLUE) are critical benchmarks for evaluation in NLP, and the labels can be leveraged to better reveal the interpretation capability of CLDA. Therefore, we need to fine-tune the PLMs for both the baseline model and our CLDA.
>
> **Q4:"The paper should discuss the literature on attention-based explanations in BERT"**
>
> Thank you for pointing us to the interesting references. We will include reference in the revision. As we discussed in the prementioned statements, our model is methodologically **different** from the existing attention-based explanations and contextual topic models.
>
> [5] Bert: Pre-training of deep bidirectional transformers for language understanding.

---

### Author Response · Authors · 2022-11-09
**Response to all the reviewers and area chairs**

Dear reviewers:

We thank all reviewers for their constructive comments. We are glad that they found our work ``"well designed"`` (YUbf), ``"novel/new"`` (c7kD, YUbf, UZmn ), ``"technically/mathematically sound"`` (iDsq, YUbf), and ``"interesting"`` (c7kD, iDsq, UZmn), and acknowledged that our approach ``"extends topic models in PLMs and uses it as interpreter and regulator"`` (c7kD, iDsq, UZmn). We address their questions below. We will also include all mentioned references (c7kD, iDsq) in the revision.

The scheme of existing methods of topic models on PLMs typically focus on learning good topics from texts with the help of contextual embeddings. And the target is to improve the coherence/diversity score compared to traditional topic models. Nevertheless, our approach aims to interpret and improve the capacity of the PLMs by learning concept-level topics from the contextual embeddings. In specific, we propose a novel probabilistic framework including the generation of word embeddings, rather than simply leverage the contextual word embeddings to learn a topic model. Therefore, our method is substantially different from existing attention-based token-level interpretations.


[1] Is Attention Explanation? An Introduction to the Debate

[2] BERTopic: Neural topic modeling with a class-based TF-IDF procedure

[3] Pre-training is a Hot Topic: Contextualized Document Embeddings Improve Topic Coherence

[4] Topic Modeling with Contextualized Word Representation Clusters

---

### Decision · Program_Chairs · 2023-01-20

**Decision:**

Reject

**Justification For Why Not Higher Score:**

The submission is not of ICLR quality.

**Justification For Why Not Lower Score:**

The submission is not of ICLR quality.

**Metareview: Summary, Strengths And Weaknesses:**

This paper proposes a topic model based on a pre-trained language model, which can serve as an interpreter and regularizer of PLM.

Strength
* A new method for combining probabilistic and pre-trained language models is proposed. There are theoretical derivations and experimental results.
* The proposed method is novel.
* The techniques are sound.

Weakness
* Experimental results need to be more convincing. The explanations given by the authors in the rebuttal could be more satisfactory.
* The motivation for the work needs to be clarified. How the proposed model can be used to meet practical needs is still being determined.
* The experiments cannot support the claims.
* The presentation needs improvements.

**Summary Of Ac-Reviewer Meeting:**

I initiated a discussion.  Three reviewers think that the submission should not be accepted.  Only one reviewer thinks that the submission can be accepted. However, the reviewer did not reply to my comment.